# Influenza A Virus H7 nanobody recognizes a conserved immunodominant epitope on hemagglutinin head and confers heterosubtypic protection

Zhao-Shan Chen[1,2], Hsiang-Chi Huang[2,3,4], Xiangkun Wang[1], Karin Schön[2], Yane Jia[1], Michael Lebens[2], Danica F. Besavilla [2], Janarthan R. Murti[2], Yanhong Ji[1], Aishe A. Sarshad [3,4], Guohua Deng[5], Qiyun Zhu [1] ✉ & Davide Angeletti [2,6] ✉

Influenza remains a persistent global health challenge, largely due to the virus' continuous antigenic drift and occasional shift, which impede the development of a universal vaccine. To address this, the identification of broadly neutralizing antibodies and their epitopes is crucial. Nanobodies, with their unique characteristics and binding capacity, offer a promising avenue to identify such epitopes. Here, we isolate and purify a hemagglutinin (HA)-specific nanobody that recognizes an H7 subtype of influenza A virus. The nanobody, named E10, exhibits broad-spectrum binding, cross-group neutralization and in vivo protection across various influenza A subtypes. Through phage display and in vitro characterization, we demonstrate that E10 specifically targets an epitope on HA head which is part of the conserved lateral patch and is highly immunodominant upon H7 infection. Importantly, immunization with a peptide including the E10 epitope elicits cross-reactive antibodies and mediates partial protection from lethal viral challenge. Our data highlights the potential of E10 and its associated epitope as a candidate for future influenza prevention strategies.

Influenza A (IAV) and B viruses are responsible for seasonal epidemics, leading to 290,000 to 650,000 human deaths globally each year[1–3]. Despite vaccination efforts, the continuous circulation of these viruses in humans, coupled with their ability to rapidly mutate, poses a significant ongoing public health threat. In addition, the current spread of H5N1 among various animal species, including cattle, is causing widespread concerns[4]. Similarly, avian influenza A (H7N9), circulating in birds and poultry, has led to over a thousand laboratory-confirmed human infections with a case-fatality rate of approximately 39% though human-to-human transmission has not been reported yet[5,6]. The H7N9 virus was first identified in China in March 2013[7–9], with its the largest outbreak during the 5th epidemic wave in 2016−2017[10,11] characterized by antigenic drift in the hemagglutinin (HA) protein. H7N9 may even possess greater pandemic potential than H5N1, highlighting the

[1]State Key Laboratory for Animal Disease Control and Prevention, College of Veterinary Medicine, Lanzhou University, Lanzhou Veterinary Research Institute, Chinese Academy of Agricultural Sciences, Lanzhou, China. [2]Department of Microbiology and Immunology, Institute of Biomedicine, University of Gothenburg, Gothenburg, Sweden. [3]Department of Medical Biochemistry and Cell Biology, Institute of Biomedicine, University of Gothenburg, Gothenburg, Sweden. [4]Wallenberg Centre for Molecular and Translational Medicine, University of Gothenburg, Gothenburg, Sweden. [5]State Key Laboratory for Animal Disease Control and Prevention, Harbin Veterinary Research Institute, Chinese Academy of Agricultural Sciences, Heilongjiang, China. [6]SciLifeLab, Institute of Biomedicine, University of Gothenburg, Gothenburg, Sweden. ✉e-mail: zhuqiyun@caas.cn; davide.angeletti@gu.se

urgency of developing effective therapeutic strategies[12]. Currently, the antiviral treatment for H7N9 are limited to neuraminidase (NA) inhibitors[13]. Vaccination remains the most effective way of preventing IAV[14] reducing the risk of illness by 40–60%[15,16]. However, it is crucial to identify conserved sites of vulnerability in both human and avian influenza viruses to develop antibody therapy or novel vaccines.

Influenza HA is the immunodominant surface glycoprotein of IAV and targets of most of the antibody (Ab) response. HA can be broadly divided into globular head and stem domains[17]. Most of the Ab response is targeting variable regions such as the canonical antigenic sites in HA head[18–21]. On the other hand, stem-specific Abs exhibit higher binding breadth and are often capable of neutralizing several IAV strains[22,23]. Indeed, cross-neutralizing stem Abs can provide a further line of defense against the virus[17,24,25]. Of note, while Abs to HA-head are mostly strain-specific, several broadly neutralizing HA-head Abs have also been identified[26–28], underscoring the importance of targeting conserved epitopes in the development of universal vaccines. Among broadly neutralizing HA-head Abs are those targeting the receptor-binding site (RBS) and the lateral patch on H1[29,30]. Yet, there is limited knowledge about lateral patch-binding Abs upon H7N9 infection[5]. In general, the characterization of the epitopes of broadly neutralizing antibodies has greatly aided the development of several universal influenza vaccine candidates[28,31,32].

Beside Abs, nanobodies[33], derived from camelids, offer a promising alternative due to the small size (15 kDa)[34], strong physical[35] and chemical stability[36], and ability to access cryptic viral sites and enhance tissue permeability[37–40]. Furthermore, nanobodies are easier to produce and are poorly immunogenic in humans[41,42]. A broadly neutralizing nanobody targeting multiple influenza subtypes could be a valuable addition to our therapeutic arsenal, particularly in the event of a new pandemic[43–45].

However, immunodominance in B cell responses often steers Ab away from these conserved regions[46], directing them to variable areas of HA head instead. Therefore, understanding the mechanisms of immunodominance is essential for refocusing Abs responses to desired targets within HA[47]. While HA immunodominance in H1N1 and H3N2 is well-characterized[18,48,49], it remains largely unexplored in H7N9, especially regarding how antigenic drift redirects B cell and Ab response to H7N9[50,51]. Indeed, absence of strongly immunodominant sites may be beneficial in allowing the Ab response to more equally spread across several targets.

In this work, starting with H7 HA alpaca immunization, we identify a broadly neutralizing and broadly protective nanobody targeting the lateral patch region of HA. Furthermore, immunization, with a peptide spanning the corresponding epitope, elicits a cross-reactive, protective response. Our results provide a potential therapeutic tool and a blueprint for designing an effective universal vaccine against influenza viruses with pandemic potential.

## Results

### Generation and characterization of H7-specific nanobodies

The H7N9 virus is a rapidly evolving virus with similar pandemic potential as H5N1[5,52], making it a particularly attractive, but challenging, target to study. Indeed, previous studies on nanobodies have primarily focused on H1N1 and H3N2 HA proteins[25,30]. We selected H7N9 virus A/Environment/Suzhou/SZ19/2014 (SZ19)[53], previously isolated in our laboratory, as a well-characterized strain with high relevance to recent outbreaks, making it an ideal candidate for studying the effectiveness of H7-specific nanobodies.

To identify SZ19 H7-specific nanobodies, we immunized alpacas intramuscularly five times with inactivated H7N9 virus (Fig. 1a). After 14 days from the last immunization, we collected peripheral blood mononuclear cells (PBMCs) and constructed a phage display library. Using SZ19 H7-HA protein as the target, we performed three rounds of bio-panning, leading to selection of 96 colonies. Among them, we

sequenced the top 20 binders and identified six unique nanobodies (A11, C11, E10, F3, H10, and H12), and expressed them with Fc tags for enhanced functionality (Fig. 1a and Supplementary Fig. 1A, highlighted in yellow). As shown by the molecular model[54] nanobodies have one single-domain, and even after linking with Fc tag, the total size is only about 40 kilodaltons (kDa) (in Fig. 1b and Supplementary Fig. 1B, C), which is approximately three times smaller than a regular Ab (150 kDa)[55]. While F3 and H10 appeared closely related, all other nanobodies, presumably came from different precursors; E10 stood out as it had the longest CDR3, of 17 amino acids, and for the presence of several negatively charged amino acids (Supplementary Fig. 1F). To test their ability to recognize folded viral HA, we performed immunofluorescence assay (IFA) as well as western blotting (WB) at different time points after infection of A549 cells (Fig. 1c, d and Supplementary Fig. 1D, E). All six nanobodies were able to recognize HA when linearized in WB (Fig. 1d and Supplementary Fig. 1E) and the cytoplasm of SZ19 virus-infected A549 cells (Fig. 1c), suggesting that they all target a linear epitope which is also present on nascent HA.

Neutralizing Abs can block viral spread by preventing infection[56], and it is indeed desirable for nanobodies to be neutralizing[57]. To test neutralization, we incubated various concentrations of nanobodies with H7N9 viruses at 37 °C for 1 h before adding them to Madin-Darby Canine Kidney (MDCK) cells. Thereafter, we measured viral replication using two methods. First, we took the supernatant after 72 h and measured 50% neutralization titer ($NC_{50}$) using hemagglutination assay (Fig. 1e). Alternatively, we waited for 20 h and subjected infected MDCK to cell-based ELISA using an NP-specific Ab (Fig. 1f). Both methods showed similar results, indicating that four nanobodies were neutralizing (A11, E10, H10, F3), while H12 and C11 were not. Calculation of 50% inhibitory concentration ($IC_{50}$) by cell-based ELISA demonstrated that A11, E10 and F3 had similar potency, while H10 had an approximately 4–7-fold better neutralization potency (Fig. 1f).

To further explore the characteristics of nanobodies, we sought to map out the nanobodies binding site and see whether they bind close to HA-head or stalk. As a first rough indication we used hemagglutination inhibition test[58]. All neutralizing nanobodies (A11, E10, H10, F3) were able to inhibit hemagglutination (Fig. 1g) with a high HI titer. Surprisingly, even the non-neutralizing nanobody H12 demonstrated HI activity, albeit at a lower level compared to the other nanobodies. Indeed, the reasons behind this discrepancy are unclear and will be elucidated in future studies. Overall, the results suggested that all nanobodies, except C11, bind to HA-head.

### Nanobody E10 exhibits cross-group binding and cross-neutralization capacity

Following the identification of four potent nanobodies capable of neutralizing the homologous H7N9 virus, we expanded our analysis to determine if any of these could also recognize and neutralize other IAV strains. HA is phylogenetically divided into two major groups: group one, including circulating H1, and group two, comprising circulating H3 and avian H7 strains[59]. Typically, antibodies targeting the HA head are strain-specific[60], however, some can recognize conserved epitopes across multiple subtypes, offering cross-group neutralization[61]. Nanobodies, due to their unique prolate (rugby ball-shaped) structure and compact variable heavy-chain domain (VHH), form a convex paratope surface[33], with higher possibility to access antigen cavities that are often hidden from conventional antibodies, increasing their potential to recognize conserved epitopes invisible to human Abs.

First, to evaluate the binding capacity of the six nanobodies, we performed ELISA on plates coated either with either UV-inactivated virus or recombinant HA proteins. We tested not only H7 HA but also A/Puerto Rico/8/1934 (PR8, H1N1) and A/Hong Kong/1968 (X31, H3N2)

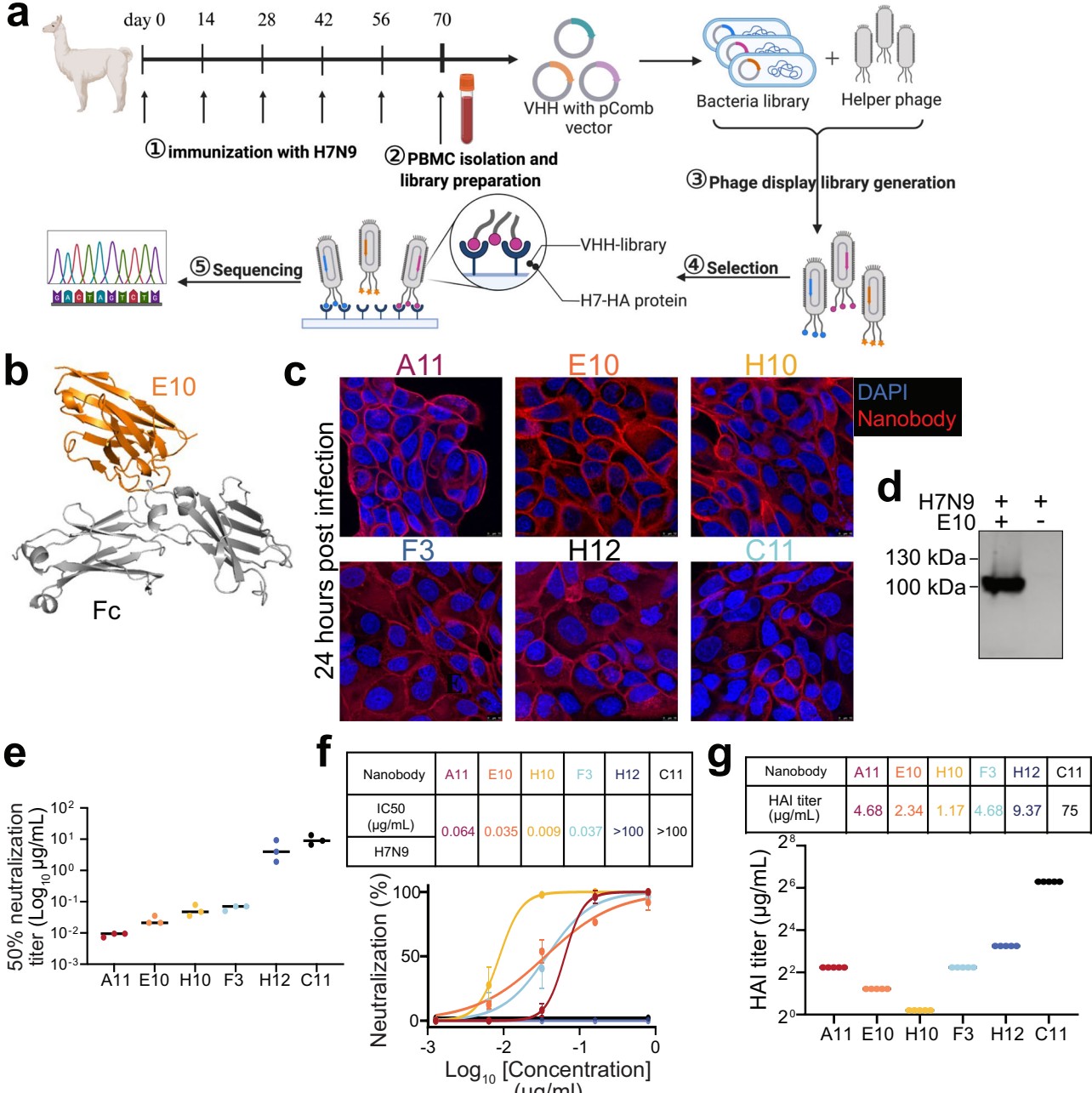

**Fig. 1 | Generation and characterization of H7-specific nanobodies. a** Schematic illustration of alpaca immunization process and subsequent nanobody selection. Alpacas were immunized with the H7N9 virus, and after four boosts, peripheral blood mononuclear cells (PBMCs) were collected to generate a phage display library. Created with BioRender.com. **b** Structural model of a nanobody-Fc created using ImmuneBuilder and AlphaFold 2. (Nanobody: orange. Fc tags: gray).
**c** Immunofluorescence assay (IFA) showing the recognition of SZ19 H7N9 virus by different nanobodies. A549 cells were infected with SZ19 (MOI = 0.1) for 24 h and stained with primary nanobodies, followed by secondary goat anti-human IgG Fc Alexa Fluor™ 488 (A11: red, E10: orange, H10: yellow, F3: blue, H12: dark blue, C11: black.). Scale bar, 10 μm. **d** Western blot (WB) analysis of A549 cells infected with SZ19 H7N9 (MOI = 1) for 24 h. Cell lysate was probed with respective nanobodies and detected using goat anti-human IgG-Fc HRP. Data are representative of at least

two independent experiments. **e** Graph showing 50% neutralization endpoint (NC$_{50}$) of different nanobodies against the SZ19 H7N9 virus. Virus was incubated with serially diluted nanobodies before cell infection. After 72 h, viral replication was measured by the ability of supernatant to hemagglutinate red blood cells (RBCs). Data are representative of at least two independent experiments. Shown are the mean values of three replicates. **f** Neutralization assay results for the nanobodies on SZ19 H7N9 virus using a cell-based ELISA. Half-maximal inhibitory concentrations (IC$_{50}$) are shown for each nanobody. Data represent the mean values ± SD from four technical replicates of three independent experiments.
**g** Hemagglutination inhibition assay (HAI) with the nanobodies against SZ19 H7N9 virus (A11: red, E10: orange, H10: yellow, F3: blue, H12: dark blue, C11: black.). HI titer shown as μg/mL. Data are the average of three independent experiments.

viruses and their respective HA proteins. The nanobodies displayed varying binding ability across different viral subtypes and HA proteins. Nanobody E10 consistently demonstrated strong binding to all viruses and HA proteins tested, suggesting broad binding capacity (Fig. 2a–f).

We further examined the neutralization capacity of E10 using the methodology described for Fig. 1e, and it demonstrated effective neutralization of H1N1 and H3N2 (Fig. 2g). These results highlighted how E10 was better than other nanobodies, demonstrating cross-

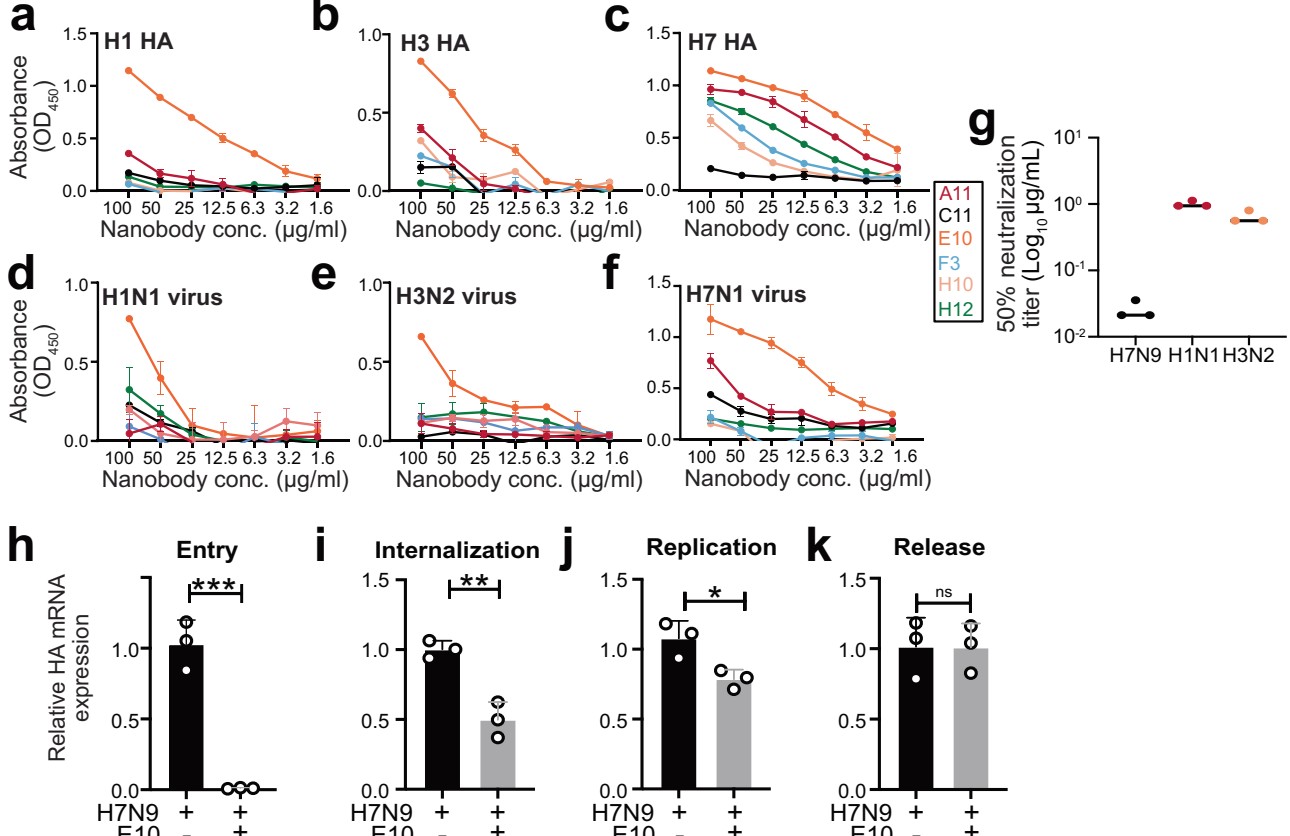

**Fig. 2 | Nanobody E10 exhibits cross-group binding breadth and neutralization capacity. a–f** ELISA binding curves showing the interaction of six nanobodies with HA protein from various influenza strains (H7, H1, and H3) and UV-inactivated influenza viruses (H7N1, H1N1, and H3N2). Each curve represents a different nanobody, color-coded for clarity. Data are presented as mean ± SD. Experiment was performed three times. **g** Neutralization efficacy of nanobody E10 against different subtypes of influenza A virus (H1N1 and H3N2). Neutralization of SZ19 H7N9 from Fig. 1e is shown for comparison. Data are representative of at least two independent experiments. Shown are the mean values of three technical replicates. **h–k** Evaluation of E10's effect on different stages of the viral infection process using SZ19 H7N9 virus (MOI = 10) at various time points. Bars represent the mean HA gene expression levels measured by qPCR ± SD. Data from three technical replicates. Statistical significance was determined using two-sided unpaired *t*-test: *p = 0.025; **p = 0.0035; ***p = 0.0005; ns not significant.

group binding and neutralization capabilities. Given these promising findings, the rest of the study focused primarily on unraveling the mechanisms behind E10's broad efficacy.

The life cycle of influenza virus can be divided into key stages: attachment, entry, internalization and fusion, genome replication, and viral release. Most antibodies targeting the HA head prevent viral entry by blocking attachment[56]. Likewise, E10 blocked attachments, as detected by HAI. We therefore wanted to determine whether other mechanisms were also involved in E10 effect. First, we tested its ability to block early entry. We pre-incubated E10 with SZ19 at 37 °C for 1 h, then added the mixture to A549 cells for 1 h to allow binding and entry (Fig. 2h). The results clearly showed that E10 efficiently blocked viral attachment and/or early entry. Next, to verify E10 ability to block viral internalization and gene replication, we followed the same protocol but added a 4 h (internalization) or 8 h (gene replication) incubation at 37 °C, after adding it to cells (Fig. 2i, j). Finally, to check whether E10 could inhibit viral release, we added the nanobody after allowing viral infection for 10 h (Fig. 2k). The most pronounced effect of E10 was on attachment and entry, with some effect on viral internalization but no influence on viral release, thus establishing its function in inhibiting attachment and entry.

## E10 treatment protects mice against homo- and heterosubtypic IAV challenge

Following the in vitro demonstration of E10's capacity to neutralize multiple influenza strains by blocking viral attachment to host cells, we proceeded to evaluate its efficacy in vivo. For prophylactic assessment, we administered 10 mg/kg of E10 to mice intraperitoneally, followed by infection, after 24 h, with $10^6$ EID$_{50}$ of SZ19 H7N9 virus (Fig. 3a). In parallel, we assessed the therapeutic potential by administering 25 mg/kg of E10 2 h post IAV infection (Fig. 3b). Mice were either sacrificed on day 3 to check for lungs pathology and viral titers in various organs, or their weight was monitored over 14 days, with a 25% weight loss used as a cutoff.

E10 treatment significantly prevented weight loss and mortality, both when administered before and after infection (Fig. 3c). In untreated mice, H7N9 spread to multiple organs, including the liver, spleen, and kidneys (Fig. 3f). However, E10 treatment completely abolished viral replication in most organs, including the lungs, with only a few mice showing viral titers in nasal tissues, indicating that E10 effectively protects various organs in vivo (Fig. 3f). As expected, therapeutic administration did not entirely prevent viral replication in the lungs, but it did reduce viral loads by 5 logs compared to untreated mice (Fig. 3f). Lung pathology analysis further confirmed the protective effect of E10, regardless of timing of administration. While H7N9 infected, untreated mice exhibited obvious lung lesions and granulocyte infiltration by day 3 post-infection, mice treated with E10, either prophylactically or therapeutically, showed normal lung tissue architecture with no visible signs of immune cell infiltration (Fig. 3g).

Given E10's cross-neutralization ability in vitro, we tested heterosubtypic protection in vivo. We used mouse adapted versions of

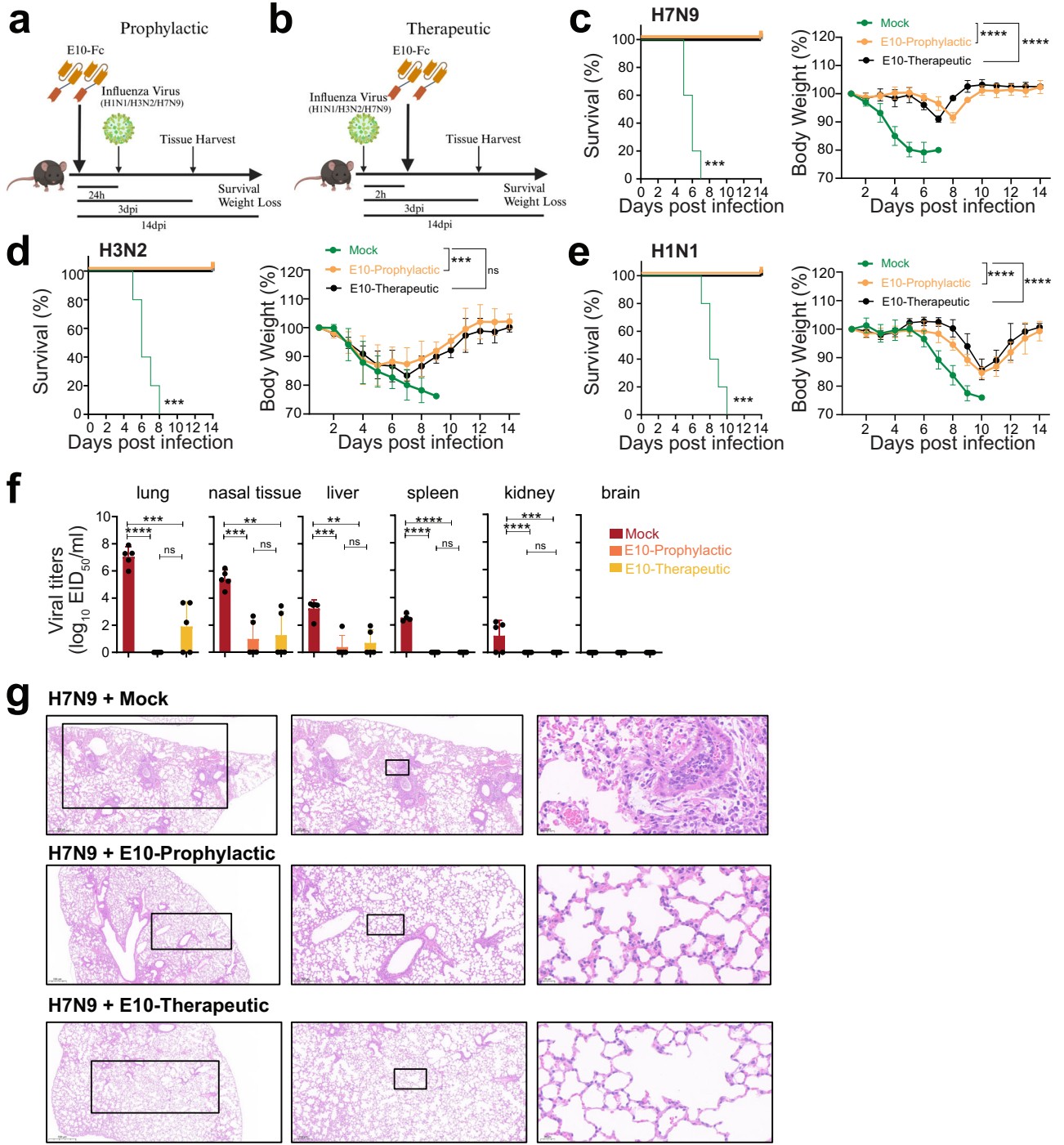

**Fig. 3 | E10 treatment protects mice against homo- and heterosubtypic IAV challenge. a, b** Schematic representation of the experimental design for in vivo studies. Mice were treated with either E10-Fc or PBS intraperitoneally (i.p.), followed by influenza virus infection (H1N1, H3N2 or H7N9), as outlined. Created with BioRender.com. **c–e** Kaplan–Meier survival curve and body weight monitoring of influenza-infected mice. Mice were treated with E10-Fc or PBS as prophylaxis or therapy i.p and weight monitored for 14 days after infection with H1N1 ($3 \times 10^3$ TCID$_{50}$, lethal dose)/H3N2 ($10^7$ TCID$_{50}$, lethal dose)/or H7N9 ($10^6$ EID$_{50}$, lethal dose). Statistical significance of the Kaplan–Meier survival curves was calculated by log rank Mantel–Cox test. (H7N9 ***$p = 0.0002$; H3N2 ***$p = 0.0001$; H1N1 ***$p = 0.0001$). Weight change data are represented as mean ± SEM from three independent experiments

($n = 15$ per group). Statistical analysis was performed using two-way ANOVA: ***$p = 0.0009$; ****$p < 0.0001$, ns not significant. **f** Viral titer, measured by EID$_{50}$, in six organs (lungs, nasal tissue, liver, spleen, kidney, brain) on day 3 post-SZ19 H7N9 infection, with or without E10 treatment, as described in (**a, b**). Data are representative of three independent experiments and shown as mean values from three technical replicates. Statistical analysis was performed using two-sided unpaired $t$-test. *$p < 0.05$; **$p < 0.01$; ***$p < 0.01$; ****$p < 0.0001$; ns not significant. **g** Representative histopathological analysis of lungs from mice on day 3 after H7N9 infection with or without E10, or MOCK treatment, as outlined in (**a, b**). Scale bar, 500 μm (left), 200 μm (middle), 20 μm (right).

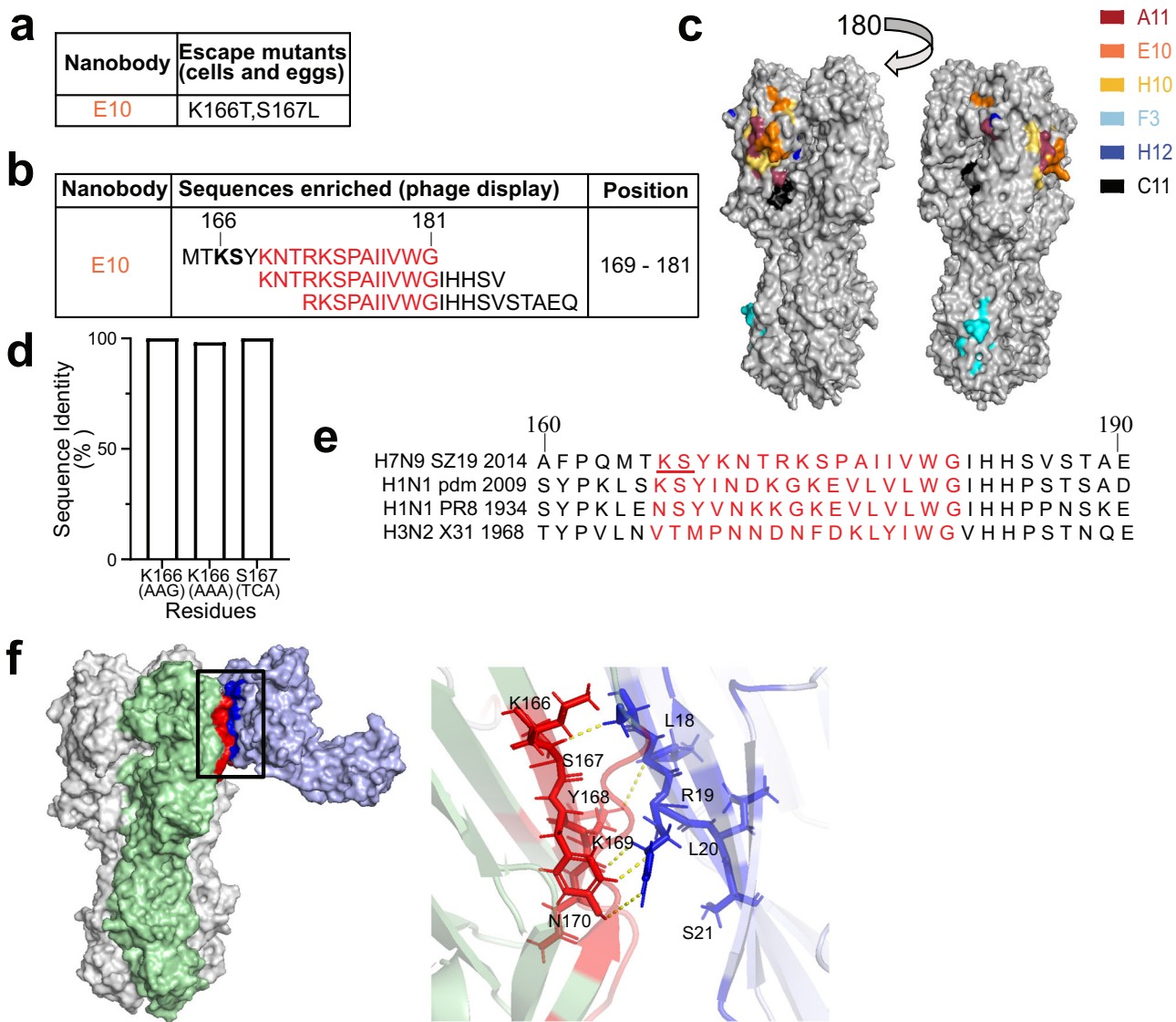

**Fig. 4 | E10 recognizes a conserved epitope located on HA-head lateral patch.**
**a** E10 escape mutations in the H7N9 virus were identified using SPF eggs and MDCK cells. Variations at key residues involved in nanobody escape are highlighted, showing the sites where resistance developed. **b** Phage display selection of E10 nanobody identified specific peptides. The red region depicts the overlapping area among recognized peptides. The bolded region corresponds to the epitope identified through escape mutations, which overlap with key binding residues.
**c** Epitope mapping of the nanobodies on SZ19 H7N9 HA protein showing the binding sites of six different nanobodies, each labeled with distinct colors (A11: red, E10: orange, H10: yellow, F3: blue, H12: dark blue, C11: black). The head domain of SZ19 H7 HA was modeled using Swiss-Model. Images were generated using Open-Source PyMOL version 2.5.0. **d** Quantitative analysis of amino acid substitutions at positions 166 and 167 across all H7N9 strains in the GISAID database. **e** Amino acid sequences flanking the E10 epitope are compared across various IAV strains, including A/Environment/Suzhou/SZ19/2014 (H7N9), A/California/07/2009 (H1N1 Pandemic), A/Puerto Rico/8/1934 (PR8, H1N1), and A/Hong Kong/1968 (X31, H3N2). Putative E10 epitope is highlighted in red, with the mutations selected by E10, at positions 166 and 167, underlined. **f** Three-dimensional (3D) model analysis of intramolecular interactions between E10-Fc and H7N9-HA obtained using ClusPro 2.0[84,85]. On the left, the HA trimer with two subunits in white and one in green; on the right, the modeled E10-Fc in light blue. The red area on HA is the interaction area, according to the model, while in dark blue is the interaction area on the nanobody. The black box indicates the area that is enlarged on the left side.

H1 and H3 IAV subtypes, belonging to two distinct phylogenetic groups. Animals were infected with either PR8 (H1N1) or X31 (H3N2), with E10 administered before or after infection and their weights monitored (Fig. 3d, e). All untreated, infected mice succumbed to infection by day 9 (H3N2) or day 10 (H1N1). However, consistent with its in vitro activity, E10 was able to convey protection against both strains, reducing mortality to zero, despite some weight loss, even in animals which received the nanobody (Fig. 3e–j). Likewise, E10 administration also significantly reduced viral titers and lung pathology in heterosubtypic infections (Supplementary Fig. 2A, B).

## E10 recognizes a conserved epitope located on HA-head lateral patch

To identify the binding site of nanobodies, we generated escape virus variants using 10-days-old specific pathogen-free (SPF) eggs (Fig. 4a and Supplementary Fig. 3A). This approach takes advantage of the error-prone polymerase of IAV to determine the Ab-epitopes[62]. Different dilutions of each nanobody were mixed with H7N9 virus and injected into the eggs. After three to four rounds of selection, hemagglutination inhibition-positive supernatants were submitted for next-generation sequencing (NGS). E10-selected escape mutants at residues K166T and S167L (H3 numbering, used throughout the

manuscript), A11-selected escapes at K140N, while H10 at residues S167L and G205E.

Furthermore, we repeated the selection process for E10 using cell culture, and obtained the same escape mutants, thus confirming its binding site (Fig. 4a and Supplementary Fig. 3A). Cell culture escape selection results for A11 and H10 were comparable to those obtained in eggs, with H10 additionally selecting for a mutation at residue S167, while H12 now selected for mutants. Furthermore, we constructed an H7 fragment library for phage display selection, utilizing 20-mer peptides overlapping by 15 amino acids. We panned phages using all nanobodies (Supplementary Fig. 3B) and identified binding sites which were in agreement with those selected in cells and egg. Of note, E10 selected for three overlapping peptides, one of them including the K166 and S167 residues (Fig. 4b) and the others in close proximity. Altogether, by combining the three different methods we mapped all the nanobodies epitopes on a molecular model of H7 HA (Fig. 4c and Supplementary Fig. 3C, D). Interestingly, E10, A11 and H10 bound in a similar, but not overlapping, region near the lateral patch of HA while the F3 epitope localized to the stem of HA. As we were unable to select escape mutants using F3, we could not confirm this mapping with other methods; furthermore, since F3 has HAI and neutralizing activity, the precise location of its epitope remains unclear and should be investigated further.

Conservation analysis of the E10 mutated residues showed that both K166 and S167 are highly conserved across H7-HA proteins (Fig. 4d), suggesting limited immune pressure on this site or lower viral fitness of escape mutants. While these residues were fully conserved in the currently circulating pandemic H1N1, some variation was observed within H1N1-HA (PR8) and H3N2-HA (X31) (Fig. 4e). However, the ~20 amino acids close to that region showed a good degree of conservation, suggesting that the nanobody footprint may also include nearby residues, as also suggested by the phage display selection. To verify this, we modeled the interaction between E10-Fc and H7-HA: indeed, most of the suggested contact surface of HA with the nanobody included conserved β-sheet structures on the lateral patch of HA (Fig. 4f). In summary, our combined mutation and epitope mapping analyses demonstrated that the broadly reactive E10 nanobody recognizes a highly conserved epitope on the lateral patch of H7 HA.

## H7-HA$_{K166T, S167L}$ mutant virus has reduced viral fitness

To confirm that the selected escape mutations were indeed part of the E10 binding site, we employed multiple experimental approaches. First, we verified E10-Fc recognition on H7-infected A549 cells and while E10 readily stained H7N9-infected cells (thereafter referred as WT virus), we could not detect any binding in H7N9$_{K166T, S167L}$-infected cells (referred to as MUT virus) (Fig. 5a). Likewise, escape mutants obtained by A11 selection failed to recognize their respective mutated virus (H7HA$_{K140N}$) (Supplementary Fig. 4A). Since E10 recognizes a linear epitope, we further verified the escape of MUT virus by WB. Here, we incubated WT or MUT virus with E10 prior to infecting cells and detected productive infection by NP antibody and HA recognition by E10. WT virus without E10 pre-incubation readily infected A549 cells and HA was detected by the nanobody. Conversely, pre-incubation with E10 blocked infectivity of WT but not MUT virus and E10-nanobody was not able to react with MUT-HA (Fig. 5b and Supplementary Fig. 4B, C). Further confirming escape, E10 was not able to neutralize MUT virus infection in vitro (Fig. 5c). Finally, we expressed H7 HA from WT and MUT viruses as recombinant HA proteins (thereafter referred to as WT-HA protein and MUT-HA protein) (Supplementary Fig. 4D) and tested nanobody reactivity. As expected, E10 lost all reactivity towards MUT-HA while F3 control nanobody recognized both proteins equally (Fig. 5d).

Finally, to test the in vivo ability of E10 to protect against MUT virus we conducted the same experiments as in Fig. 3 and determined its prophylactic and therapeutic activity upon lethal viral challenge (Fig. 5e and Supplementary Fig. 4E). Again, mice were either sacrificed

on day 3 to check for lung pathology and viral titers in different organs, or their weight monitored for 14 days (Fig. 5f, g and Supplementary Fig. 4E). The in vivo results confirmed the loss of efficacy of E10 on the MUT virus, cautioning that an escape to this broadly neutralizing nanobody is possible. However, MUT virus showed a slightly decreased pathogenicity in vivo, including a longer survival and lower lung viral titer, when compared to WT. These results suggest that MUT may have a less virulent phenotype, thus raising hopes for the targeting of this epitope. To confirm this, we compared the in vitro growth kinetics of WT and MUT viruses: WT grew rapidly and reached peak titer by 36 h post infection where it killed out most cells. Conversely MUT virus grew slower and to a peak over two logs lower than WT (Fig. 5h), demonstrating a reduced viral fitness. Overall, while the emergence of escape mutants in response to E10 nanobody is possible, their viral fitness was reduced both in vitro and in vivo.

## E10-epitope is immunodominant upon H7-IAV infection

Little is known about H7 B cell immunodominance. Previous research suggested that single amino acid differences may alter the establishment of broadly neutralizing B cells[63]. Additionally, studies on immunodominance in humans have shown that antibody responses can be highly focused on specific epitopes, and that a single amino acid mutation can allow the virus to escape immunity in some individuals[48,64–66]. We reasoned that with our current tools we would be able to dissect the immunodominance of the B cell responses to the E10 epitope upon infection.

To this end we infected mice with either WT or MUT virus and we carried out B cell staining of lungs, the draining mediastinal lymph node (mln) and the spleen by flow cytometry[67]. Beside classical markers to distinguish germinal center (GC) and memory (MBC) B cells, we also included a multiple HA staining to define epitope specificity (Fig. 6a, c and Supplementary Fig. 5A). All cells were stained with WT-HA labeled with two distinct fluorophores and MUT-HA with other two fluorophores: for WT-infected mice, we gated first on WT-HA double positive, as this was the only protein seen by the animals. Thereafter we used the staining with the MUT-HA to discriminate how many B cells were specific for the E10 epitope (Fig. 6b, d): indeed, if WT-HA positive cells lost the recognition when stained with MUT-HA this suggested a specificity for the E10 epitope. Likewise, for mice infected with MUT virus, we first gated on MUT-HA and thereafter only the ones not recognizing the WT-HA were defined as specific for the mutated E10 epitope (Fig. 6b, d).

Surprisingly, ~50% of the HA-directed GC and almost 100% of the MBC response in WT-infected mice was specific for the E10 epitope in mln (Fig. 6e), lungs and spleen (Supplementary Fig. 5B–E). This is a case of immunodominance where the response was very focused; indeed, the majority of the B cells was responding to a very small antigenic surface of H7 HA. Importantly, we did not detect any gross differences in GC size between WT and MUT after infection with the two viruses (Supplementary Fig. 5F). However, in mice infected with MUT virus, this immune focusing was lost, with only approximately 10% of B cells recognizing the mutated E10 epitope (Fig. 6e). The results here suggest a very strong immune focusing on the E10 epitope upon WT virus infection, which was however lost after nanobody-driven mutations.

To verify whether the immunodominance in WT-infected mice extended to the antibody secreting cells (ASC) and the Ab compartments, we performed ELISPOT and ELISA. Consistent with B cell results, ASC immunodominance was very pronounced in WT-infected but not in MUT-infected mice (Fig. 6f), a result that was further confirmed when analyzing serum Abs (Fig. 6g and Supplementary Fig. 5G). Here, more than half of the Ab reactivity was lost when testing WT-infected sera on the MUT HA protein, confirming the strong immunodominance of the E10 epitope. In our study, we have not investigated the immunodominance to this epitope in different animal species, indeed existing data on species specificity of antibody immunodominance is scarce and contradictory[66,68]. However, it is

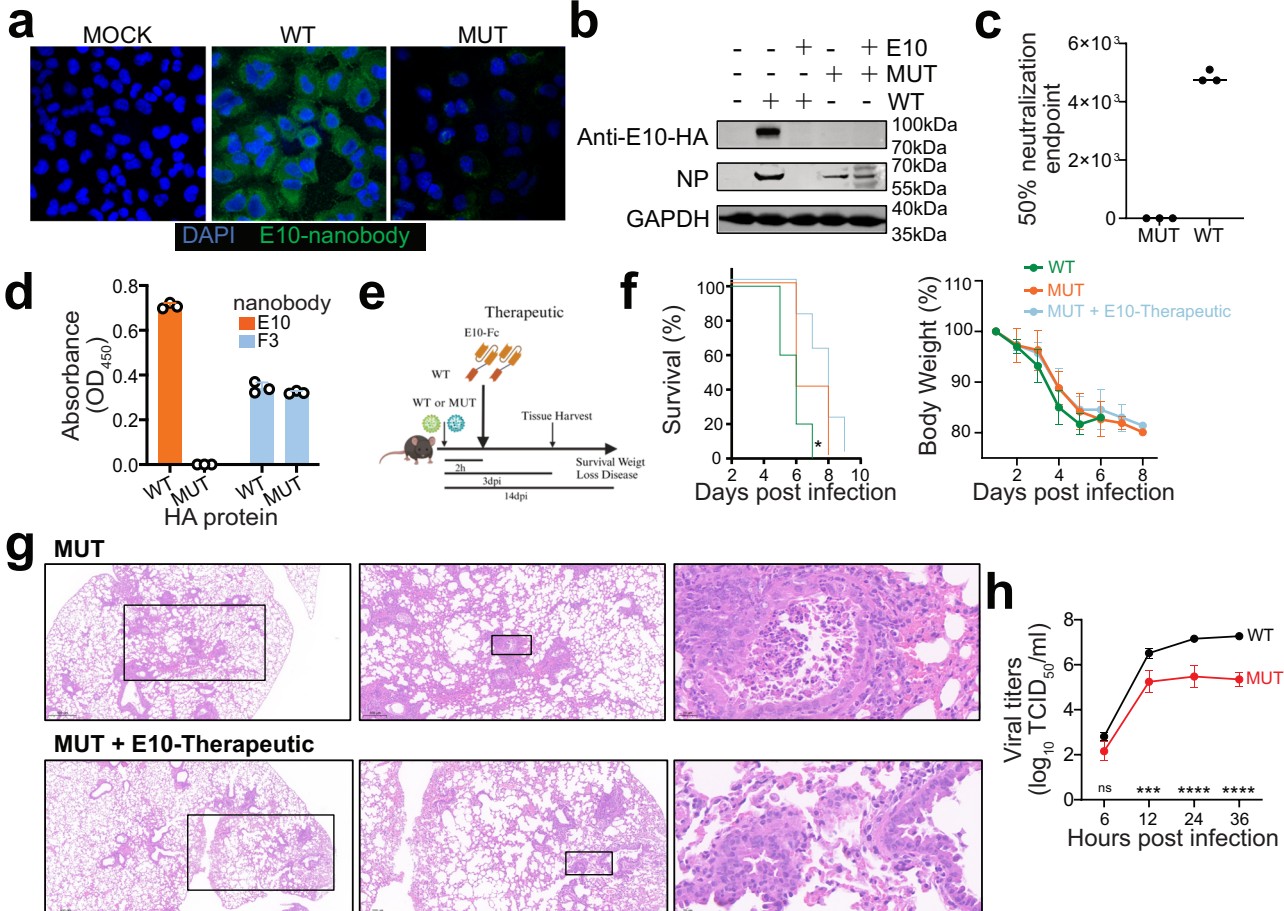

**Fig. 5 | H7-HA$_{K166T, S167L}$ double mutant escapes E10 recognition but has lower viral fitness. a** IFA of A549 infected with wild-type (WT) or mutant (MUT) H7N9 virus (MOI = 0.1) at 24 h post-infection. Cell was stained with E10 as the primary antibody, followed by secondary goat anti-human IgG Fc Alexa Fluor™ 488. Nuclei were counterstained with DAPI (blue), and NP nanobody detection is shown in green. Data are representative of at least two independent experiments. **b** WB analysis of A549 cells infected with WT or MUT H7N9 viruses, with or without E10 pre-incubation, showing detection of NP and E10. Data are representative of at least two independent experiments. **c** Graph showing the 50% neutralization endpoint of E10 against WT and MUT viruses, measured by the ability of virus progeny to hemagglutinate red blood cells (RBCs). Data are representative of at least two independent experiments. Shown are the mean values of three technical replicates. **d** ELISA detection of WT-HA and MUT-HA hemagglutinin (HA) by E10 and F3 nanobodies. Bars represent mean ± SD. Data are representative of at least two independent experiments. Shown are the mean values of three technical replicates.

**e** Experimental setup for the studies shown in (**f**, **g**). Created with BioRender.com. **f** Kaplan–Meier survival curve and body weight analysis of influenza-infected animals. Mice were treated intraperitoneally (i.p.) with either E10-Fc or PBS, as depicted in (**e**), and their weight was monitored for 14 days following infection with WT (10⁶ EID₅₀, lethal dose) or MUT virus (10⁶ EID₅₀). The statistical significance was calculated by log rank Mantel–Cox test for survival curve, p = 0.04. Weight change was monitored, and each graph is three experiments; n = 15; symbols represent means ± SEM. **g** Representative histopathological analysis of mouse lungs at day 3 post-infection with MUT virus, with or without E10 administration as outlined in (**e**). Scale bar, 500 μm in the left row, 200 μm in the middle row, 20 μm in the right row. **h** Growth kinetics of WT and MUT viruses in MDCK cells. Supernatants were collected at 6-, 12-, 24-, and 36-h post-infection (MOI = 0.001) and titrated by TCID₅₀. Virus titers are presented as mean ± SD from three independent experiments. ***p = 0.001, ****p < 0.0001 (two-way ANOVA followed by Sidak test).

plausible to think that this site may be under selection pressure in its natural host, and despite that, it continues to show conservation, possibly because of the lower fitness of MUT virus (Fig. 5h).

## H7-HA$_{166-186}$ peptide immunization confers partial in vivo protection from lethal H7N9 infection

Having verified the ability of E10 to provide heterosubtypic protection and the relative stability of its epitope, we decided to verify whether immunization with peptide containing the E10-binding motif would be sufficient to provide protection in animals. Therefore, we expressed the 21 mer peptide corresponding to H7HA$_{166-186}$ sequence (KSYKNTRKSPAIIVWGIHHSV) linked with OVA in the C-terminal to increase immunogenicity and availability of helper T cell epitopes. The peptide corresponds to two β-sheet in HA1 (Fig. 7a). We immunized mice thrice, collected serum 6 days after the third immunization and challenged the mice with a lethal dose of H7

14 days after the last vaccination (Fig. 7b). Serum Abs from mice vaccinated with the peptide demonstrated broad reactivity, recognizing not only full length H7 HA but also H3 and H1 HAs, in line with E10´s binding profile (Fig. 7c and Supplementary Fig. 6). The binding capacity was confirmed also by WB analysis of viral lysate (Fig. 7d). To check the ability of serum to recognize live virus, we tested sera from peptide-immunized mice for their ability to block hemagglutination: similar to E10, peptide-immunization elicited Abs capable of blocking hemagglutination of viruses belonging to both groups (Fig. 7e). Finally, peptide immunization slightly delayed weight loss and protected 60% of the vaccinated mice upon H7 viral challenge, confirming the central role of this epitope for protection (Fig. 7f).

Overall, peptide immunization, despite its notorious limitations[69], was able to elicit significant levels of cross-reactive Abs in mice and to afford partial protection against lethal viral challenge thus highlighting the potential and relevance of E10 epitope as vaccine target.

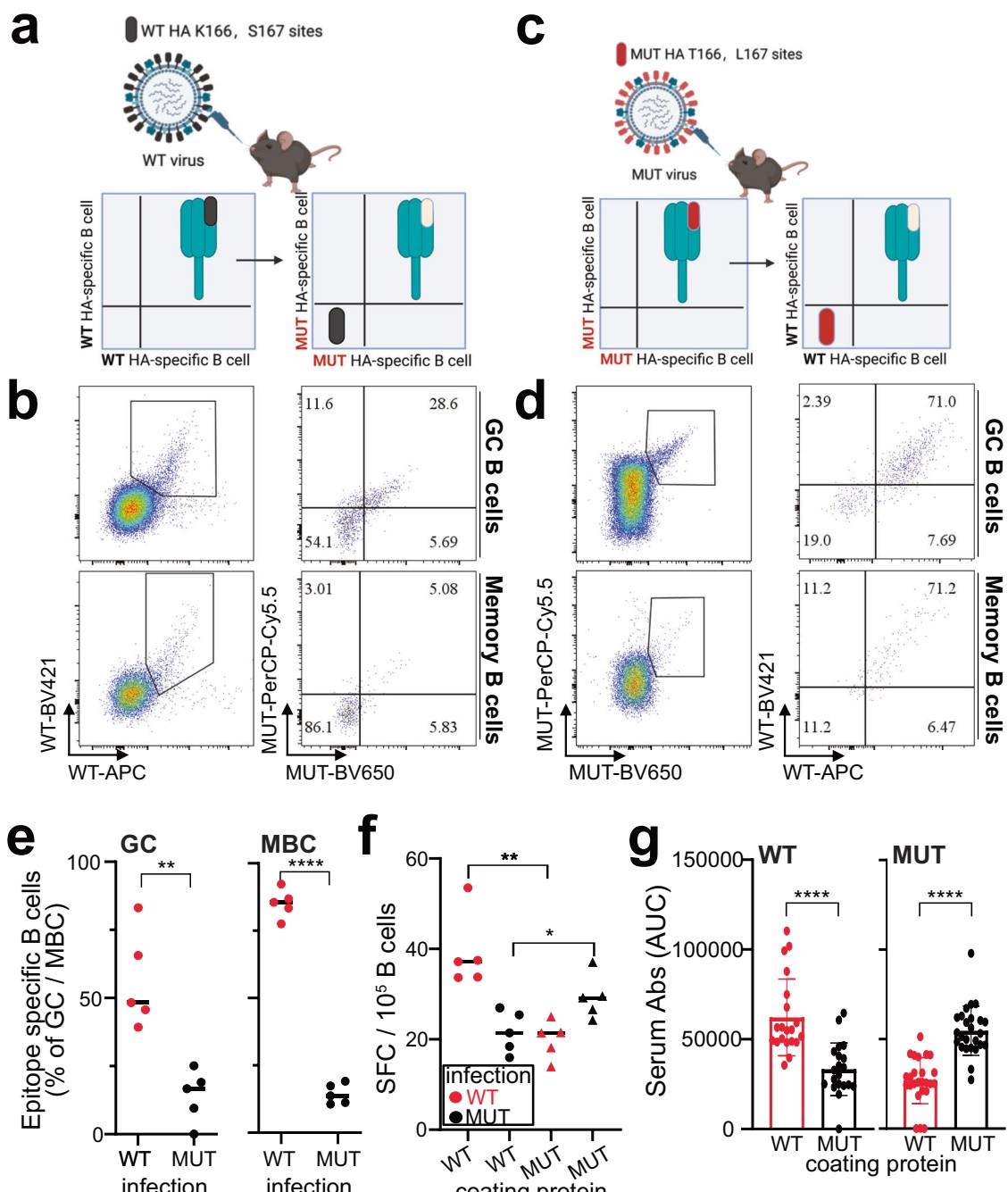

**Fig. 6 | E10-epitope is immunodominant upon H7-IAV infection. a, c** Schematic illustration of epitope identification by flow cytometry. Created with BioRender.com. **b** Representative flow cytometry gating of GC and MBC for epitope identification in mln of WT-infected mice 14 days post-infection. GC B cells were gated as live CD3⁻ B220⁺ IgD⁻ IgM⁻ GL7⁺ CD38⁻, MBC as live CD3⁻ B220⁺ IgD⁻ IgM⁻ GL7⁻ CD38⁺. **d** Same as in (**b**) but for MUT-infected mice. **e** Quantification of epitope-specific MBC and GC B cells in mln of WT- and MUT-infected mice at 14 days post-infection. Two independent experiments with 5 mice each. Bars represent SEM; statistical analysis was performed using two-sided unpaired *t*-test.

**p = 0.0016, ****p < 0.0001. **f** Quantification of ASC by ELISPOT, plates were coated with WT-HA and MUT-HA, and ASC were quantified in mln of WT vs. MUT infected mice. Spot-forming cells were normalized to 10⁶ cells. Data from five mice per group. Bars represent SEM, statistical analysis was performed using two-sided unpaired *t*-test. *p = 0.034, **p = 0.0017. **g** Serum reactivity to WT-HA and MUT-HA in WT- and MUT-infected mice at 14 days post infection. Area under the curve (AUC) quantification is depicted. Data represent four independent experiments with five mice each (*n* = 20). Bars represent mean ± SEM; statistical analysis was performed using a two-sided unpaired *t*-test. ****p < 0.0001.

## Discussion

H7N9 remains a serious public health threat with frequent human infections, which are often lethal[12]. Furthermore, the risk of virus adaptation, gaining human-to-human transmission ability, in a naïve population, is a looming danger. Developing novel therapeutic tools to treat zoonotic infections but also discovering sites of vulnerability on

this virus is imperative. Compared to mAbs, nanobodies represent a promising new avenue for preventing and treating influenza due to their superior characteristics and distinct binding capacity compared to conventional antibodies[40]. Here, we isolated a broadly neutralizing nanobody, E10, capable of binding multiple strains of influenza A virus. To enhance the therapeutic potential of E10, we added human Fc

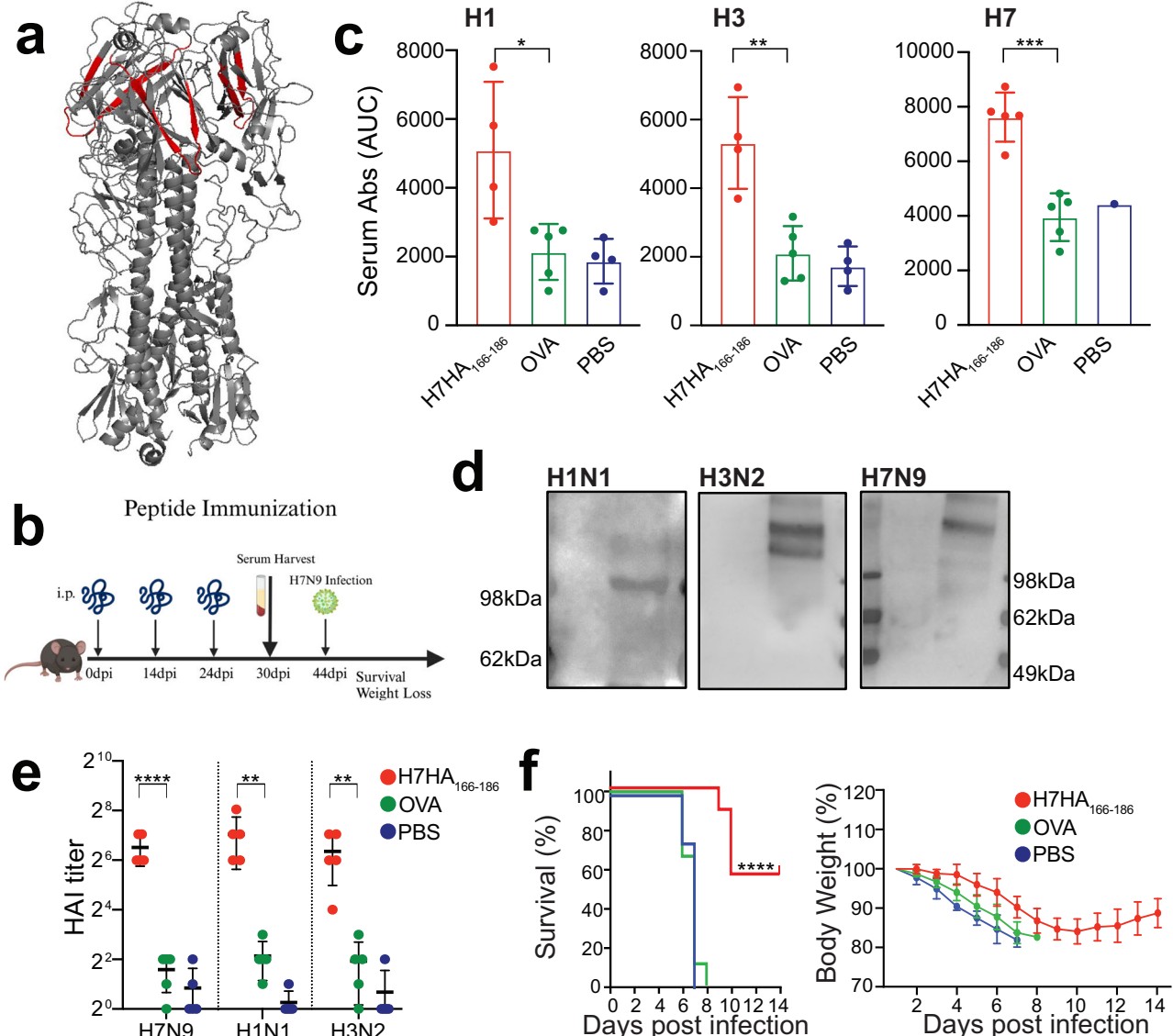

**Fig. 7 | H7HA$_{166-186}$ peptide immunization confers partial protection from lethal H7N9 infection. a** Schematic illustration of H7-HA$_{166-186}$ peptide localization (shown in red) within the HA protein timer (gray). **b** Schematic timeline of peptide immunization protocol. Created with BioRender.com. **c** Serum binding analysis to H1, H3 and H7 HA proteins from mice immunized with H7HA$_{166-186}$ peptide, OVA peptide and PBS. Data are shown as area under the curve (AUC) quantification of the ELISA curves. Representative of two independent experiments with 4–5 mice per group. Bars represent mean ± SEM, statistical analysis was performed using a one-way ANOVA followed by Tukey's multiple comparison test; shown is only the difference between H7HA$_{166-186}$ peptide vs. OVA groups. $^*p = 0.014$, $^{**}p = 0.0012$, $^{***}p = 0.0005$. **d** WB analysis of HA protein recognition from MDCK cells infected with H1N1, H3N2, or H7N9 viruses, using serum from

H7-HA$_{166-186}$ peptide-immunized mice. First antibody: serum from H7HA$_{166-186}$ peptide-immunized mice, second antibody: anti-mouse IgG. **e** Graph showing Hemagglutination inhibition assay (HAI) titer of immunized serum against H1N1/H3N2/H7N9 viruses. Representative of two independent experiments with 4–5 mice per group. Bars represent mean ± SEM, statistical analysis was performed using a one-way ANOVA followed by Tukey's multiple comparison test; shown is only the difference between H7HA$_{166-186}$ peptide vs. OVA groups. $^{****}p < 0.0001$; H1N1 and H3N2: $^{**}p = 0.003$. **f** Weight loss and survival curves of mice immunized with H7HA$_{166-186}$ peptide. Mice ($n = 10$ per group, two independent experiments) were infected with H7N1 (1 TCID$_{50}$, lethal dose) at day 42, after being immunized three times with H7HA$_{166-186}$ peptide (see **b**). $^{****}p < 0.0001$.

fragment, improving its structural similarity with human antibodies and allowing it to trigger effector functions. The neutralizing mechanism of E10 involves blocking viral attachment of the HA protein. E10 demonstrated broad in vitro and in vivo activity, providing neutralization and conferring protection against group 2 (H7, H3) and group 1 (H1) strains.

We utilized multiple methods to identify E10 binding site and pinpointed the area around residues K166 and S167 as its binding site. This area included two conserved-beta sheets and is part of the "lateral patch" of HA. Because of antigenic drift, the HA head domain is highly variable among different strains[70]. However, a few regions in the head

have gained traction as potential targets of broadly neutralizing Abs. One is the receptor binding site (RBS)[71], with Abs targeting this area mostly interacting with highly conserved residues of the RBS by inserting their HCDR3[60]. In addition, lateral patch" binding antibodies have been receiving more and more attention[5,27,29,30,72,73]. For instance, a mAb, CL6649, can recognize the lateral patch and binds most of the H1N1 viruses ranging from 1977 (seasonal) to 2012 (pdm2009)[29]. Furthermore, a recent study reported identification of two human H7N9 mAbs targeting the lateral patch on HA head[5]. These could bind to several H7 strains. Importantly, none of these Abs are able to cross-react between different HA groups or even distant types, highlighting

the distinctive properties of E10 nanobody. It should be noted that these studies, using human mAbs, tested them at concentrations up to 10 µg/ml while in our experiments we observe cross-reactive binding to H1 and H3 at higher concentrations (>10 µg/ml for binding to recombinant HA and >25 µg/ml for binding to viruses). In general, the potency, neutralizing and protective effect of E10 nanobody is higher against H7, as expected. Nevertheless, the binding is specific, as it is observed only for the E10 nanobody, and strategies to increase binding affinity may be valuable to increase potency against heterologous HA. Indeed, cross-group recognition and neutralization is a rare but desirable feature of anti-HA Abs. Possibly, thanks to the unique features of alpaca nanobodies we increased the likelihood of targeting conserved epitopes in a unique way.

Moreover, lateral patch polyclonal Abs are also induced by pandemic H1 vaccination, though it remains unclear whether these antibodies can cross-react with other influenza subtypes[30]. From the same study, the authors isolated human mAbs and identified a crucial, conserved Y-x-R or Y-R motif within the H-CDR3 region[30]. In addition, these mAbs mostly showed VH3-23/Vk1-33 usage[22,30]. Interestingly, our E10 nanobody possesses the "YCSFR" sequence in its CDR3, which is absent in other nanobodies we have identified.

In our study we also demonstrate that peptide immunization is able to generate cross-reactive Abs that can block hemagglutination in vitro and partially protect in vivo. The partial in vivo protection should not discourage as peptide immunization is notoriously poor at eliciting Abs[69], it however provides an excellent blueprint for the immunogenicity and potency of this site which could be exploited in future studies with more advanced computational immunogen designs[74].

With H5N1 infections on the rise, research on animal viruses is an urgent priority and H7N9 presents a significant pandemic risk. Here, besides the identification of the nanobody we also tested basic immunodominance characteristics of the virus. Immunodominance for H7 has not been extensively studied and defining it is not a mere academic exercise but may inform on virus adaptation and mutation potential. Indeed, immunodominance has been suggested as potential driver of antigenic drift[20]. Several studies have now pointed out that certain numbers of individuals have a very focused Ab response, and these may be driving antigenic drift[20,48,64–66]. Generally, in mice, immune response to IAV is broad and H1N1 virus require over 10 mutations to fully escape the serum of infected animals[50]. Here, we show that mutations in the E10 epitope almost completely abolish HA recognition by GC and MBC in mice. Despite this, the residues remain conserved among most of the circulating H7 strains encouraging potential therapeutic use. This suggests either a different immunodominance pattern with low immune pressure in avian hosts or poor fitness of the escape mutants. Whether immunodominance in different hosts is conserved is debated[66,68], but some data points towards a conserved hierarchy of the response between avian and mammal hosts[68]. Similarly, the ease of isolation of mAbs targeting a similar site in humans[5] suggests this epitope to be under constant immune pressure. However, our in vivo and in vitro data suggest this epitope to be needed to maintain a strong viral fitness and therefore, while escape mutants may emerge, they are unlikely to be able to compete with WT virus. Obviously increased immune pressure may favor escape in this site, as demonstrated for other conserved Ab epitopes[75–77], however, in addition to showing lower fitness, the MUT virus showed also increased epitope spread after infection, favoring a broader immune response. This is yet another good news for the potential use of E10 as therapeutic nanobody or for the targeting of the epitope with vaccine constructs.

Despite the limitations in our experimental setup, such as the lack of definitive structural determination by cryo-EM and precise affinity measurement by Biolayer Interferometry (BLI), which could have provided more detailed insights into the antibody-HA interaction and affinity, we successfully mapped the binding area of E10 as a conserved β-sheet on the HA surface. Furthermore, while the H1 and H3 strains tested hereby are not currently circulating, the epitope remains relatively conserved even in contemporary viruses. Overall, E10 may add to our arsenal of treatments for zoonotic infections and its epitope and associated peptide may provide a blueprint for future universal influenza vaccine development.

## Methods

### Inclusion and ethical statement

All the mouse studies were performed according to ethical permits 1666/19, 2230/19 and 38/23 granted by the Gothenburg Regional Animal ethics committee, as well as the Guide for the Care and Use of Laboratory Animals of the Ministry of Science and Technology of the People's Republic of China. Experiments involving H7N9 avian influenza viruses were performed in a biosafety level 3 laboratory, approved by the Chinese Ministry of Agriculture and Rural Affairs. The alpaca and virus protocols were individually approved by the Committee on the Ethics of Animal Experiments of the Lanzhou Veterinary Research Institute (LVRI) of Chinese Academy of Agricultural Sciences (CAAS) and Harbin Veterinary Research Institute (HVRI) of the Chinese Academy of Agricultural Sciences (CAAS). Details regarding the facility and biosafety measures have been previously reported[78].

### Animals

Mice, Female C57BL/6 mice, aged 8–12 weeks, were purchased from Janvier, France, and housed in a specific pathogen-free (SPF) facility at the Experimental Biomedicine Unit, University of Gothenburg. Mice were house at 20–23 °C, 45–55% relative humidity with 12-h light/dark cycles, food and water ad libitum. Additional female C57BL/6 mice were obtained from SPF (Beijing) Biotechnology Co., Ltd. in China and housed under similar conditions. Alpaca, male, aged 2.5 years, were provided by a local farm in Gansu Province, China.

### Cells, viruses and plasmids

HEK293T (American Type Culture Collection, ATCC, CRL-3216), MDCK (ATCC, CCL-34), MDCK-SIAT1 (gifted by Dr. Jonathan Yewdell at NIH) cells were cultured in DMEM (Gibco, C11995500BT) with 10% (vol: vol) FBS (Gibco, 10,270–106) and penicillin-streptomycin (Gibco, 15,140,163). A549 cells (ATCC, CCL-185) were maintained in Kaighn's modified Ham's-F12 medium (Gibco, C11330500BT) with 10% FBS and 0.01% penicillin-streptomycin. All other cells were cultured and maintained at 37 °C with 5% $CO_2$. HEK293F (ATCC) was grown in Freestyle 293 Expression Medium (Gibco, 12338018) with 0.01% penicillin-streptomycin and cultured at 37 °C with 8% $CO_2$ at 125 rpm. All cells were regularly tested for mycoplasma. No commonly misidentified cell lines were used in this study.

A/Environment/Suzhou/SZ19/2014(H7N9)(SZ19) was isolated and stored in our laboratory, A/Puerto Rico/8/1934 (PR8, H1N1) was stored in our laboratory[53]. A/Puerto Rico/8/1934 (PR8, H1N1), influenza A virus A/Hongkong//1968 (X31, H3N2) was stored in our laboratory in Gothenburg. H7N1 virus, H7N1-MUT virus (HA comes from SZ19, others come from PR8) were rescued in our laboratory in Gothenburg. The H7-HA and H7-HA$_{K166T, S167L}$ were constructed containing the H7 ORFs: one for the H7N1$_{K166, S167}$ and another for the H7N1$_{K166T, S167L}$ mutated influenza A/Environment/Suzhou/SZ19/2014/H7N9 strains, labeled as WT and MUT, respectively. The cloning strategy involved double-digestion of the pDZ plasmid with EcoRI-HF (NEB, R3101S) and XhoI (NEB, R0146S), followed by In-Fusion cloning (Takara Bio, 638948) with three PCR-amplified fragments, containing the H7 ORF and its flanking regulatory regions. Primer sequences are shown in Table S1.

### Sequencing

Constructed plasmids were sequenced and analyzed by Eurofins and Tsingke Biotechnology Co., Ltd.

## Alpaca immunization and phage display selection

Immunization: 2 years and half male alpacas were immunized intramuscularly five times, with 14 days interval, with inactivated (0.06% formaldehyde) and purified H7N9 virus. The first immunization used 300 µg of antigen in 300 µl PBS + 700 µl Freund's Complete Adjuvant (FCA, Thermo, 77140) followed by four subsequent immunizations with 200 µg of antigen in 200 µl PBS + 800 µl Freund's Incomplete Adjuvant (Thermo, 77145).

Indirect ELISA was used to measure the IgG titer against H7N9, initiating phage display construction when the titer exceeded 1:64,000.

Phage display: after 14 days from the last immunization, peripheral blood mononuclear cells (PBMC) were isolated from whole blood of alpaca, and RNA was extracted to synthesize cDNA. VHH fragments were amplified via nested PCR using specific primers (Supplementary Table 1), then cloned into the pComb phage vector and transformed into E. coil SS320 competent cells. The positive clones were sequenced, and the diversity of the antibody library analyzed with the calculated antibody library capacity, the remaining culture was transferred to Amp-Kan medium for expansion culture, and PEG/NaCl was used to concentrate to obtain the original antibody library. ELISA plates were coated with inactivated purified H7N9 at 1 µg/well, 1% polyvinyl alcohol (1% PVA) was set as a no-antigen control, 100 µL of the original antibody library was added to each well, incubated at room temperature for 2 h, and the solution was discarded; 0.1 mol L$^{-1}$ HCl was added for elution for 5 min, and an equal volume of Tris-HCl was used for neutralization; the eluate was added to logarithmic phase NEB5aF' bacteria, cultured at 200 r/min, 37 °C for 1 h, and M13 assisted phage rescue for 1 h. An appropriate amount of bacterial solution was taken for 10-fold dilution and titrated on the plate; the remaining bacterial solution was cultured overnight. Enrichment and panning were respect three times and selected by ELISA with anti-mouse M13 antibody. Then select the positive bacteria and express the nanobodies to pcDNA13.1-Fc vector and transform to DH5α competent cells. After three rounds, 96 colonies were selected, and the eluted phages were further characterized for binding by indirect-ELISA. Primer sequences are shown in Table S1.

## Nanobody production and characterization

For protein production, nanobodies were expressed in HEK293F cells maintained in Expi Expression Medium (Gibco, 12338018), followed by transfection using the ExpiFectamine™ 293 Transfection Kit (Gibco, A14525). After 5 days, culture supernatants were filtered and purified using Protein G columns on an FPLC ÄKTA start system.

## Confocal microscopy

HEK293 and A549 cells ($5 \times 10^5$ cells/well) were plated on coverslips. Cells were allowed to attach for 8 h and were left uninfected or were infected with H1N1/H3N2/H7N9 IAV (MOI = 0.1) for 24−36 h in serum free media. Cells were fixed with 4% paraformaldehyde for 20 min at room temperature and washed with PBS three times. Cells were permeabilized with 0.1% Triton X-100 in PBS for 10 min, then blocked with 5% skimmed milk for 1 h. Then cells were incubated with nanobodies as the primary and anti-human IgG (Fc specific)-488 (Invitrogen, A55747) as secondary antibodies and DAPI (Beyotime, C1002). Leica microscope (TCS SP8) were used to observe the strained cells with a 100x oil objective.

## Western blotting

Protein samples (5−20 µg) were separated into 4−12% Bis-Tris gels, transferred to Nitrocellulose membrane (Cytiva, 1060000). Proteins of interest were analyzed by hybridization with their corresponding antibodies (anti-GAPDH (ab181602), anti-NP (Sino Biological, 11675-V08B)), horse anti-mouse IgG (Vector laboratories, ZK0403), Mouse Anti-Human IgG Fc Antibody (GenScript, 50B4A9) and visualized by chemiluminescence using Thermo Scientific SuperSignal West Dura Extended Duration Substrate (Thermo Fisher, 34076).

## Neutralization assay

Serial dilutions of nanobodies (start from 100 ng/µL) or mouse serum were mixed with 100 TCID$_{50}$ of virus (H7N9/H3N2/H1N1) for 1 h at 37 °C. The mixture was then added to cultured MDCK cells in 96-well plates for 1 h. After 3 washes with PBS we incubated with DMEM containing 1 µg/ml TPCK-treated trypsin and 0.01% penicillin-streptomycin at 37 °C. To define neutralization we used two methods: (1) After 20 h, viral infection of cells was quantified by indirect ELISA by using a nanobody made in house against the nucleoprotein (NP) of influenza A virus. The final concentration of nanobody that reduced infection to 50% (IC$_{50}$) was determined using GraphPad Prism 6 software. (2) In alternative, after 72 h of incubation, we harvested the supernatant and added it to 1% chicken red blood cells (RBC) to determine the presence of virions in supernatant and calculated the 50% neutralization titer using the Kärber formula. Log50 % neutralizing titer = $L - d (s - 0.5)$, where $L$ is the log of the dilution factor, $d$ is the log difference between the dilution factors, and $s$ is the sum of the proportion of positive wells for all dilutions.

## Recombinant hemagglutinin production and testing

Recombinant HA from A/Environment/Suzhou/SZ19/2014(H7N9) (SZ19), A/Puerto Rico/8/1934 (PR8, H1N1), Hongkong/1968 (X31, H3N2) and SZ19-HA$_{K166T, S167L}$ were expressed and purified as in ref. [79]. Briefly, proteins were expressed in 293F cells and purified by affinity chromatography followed by size exclusion.

## Enzyme-linked immunosorbent assay (ELISA)

For serum binding test, 96-well plates were coated with recombinant protein (H7-HA (WT-HA) or H7-HA-K166T, S167L (MUT-HA)) at 1 µg/ml in PBS and incubated overnight at 4 °C. Wells were blocked with 100 µL 2% BSA in PBS for 1 h at RT. Plates were washed three times with PBS + 0.05% Tween and incubated with two-fold serially diluted sera (starting from 1:100) in PBS with 0.05% Tween for 1.5 h at RT. After three washes, anti-mouse Igκ (Sino Biological, 68077-R008-H) at a 1:1000 dilution was added, and plates were incubated for 1 h at RT. After three washes, plates were developed using TMB (Thermo Fisher, catalog: 34029) for 5 min at RT and subsequently blocked with H$_2$SO$_4$. The absorbance was measured with a TECAN Sunrise absorbance microplate reader (catalog: 16039400) at 450 nm.

For nanobody binding test, we coated 96-well plates with different strain of UV-inactivated virus (H1H1/H3N2/H7N1) or different strain of HA recombinant protein (H1N1-HA/H3N2-HA/H7N9-HA) at 1 µg/ml in PBS and incubated overnight at 4 °C. Wells were blocked with 100 µL 2% BSA in PBS for 1 h at room temperature. Plates were washed three times with PBS + 0.05% Tween and plates were incubated with two-fold serially diluted nanobodies (starting from 100 ng/mL) in PBS + 0.05% Tween for 1.5 h at RT. After three-time washes, mouse anti-human IgG Fc Antibody (Sino Biological, SSA001) at 1:6000 dilution was added, and developed using TMB as above.

## Hemagglutination assay

For virus rescue detection, the hemagglutination assay was performed using 1% chicken RBC suspension to confirm virus rescue and neutralization titer. Briefly, 50 µL virus supernatant was used to make 2× serial dilutions in a round-bottomed 96-well plate. The diluted virus supernatants were mixed with 25 µL of 1% chicken RBC in each well and incubated at room temperature for 1 h before the results were inspected.

## Hemagglutination inhibition (HAI) assay

H7N9 viral stocks were standardized to 4 HAU units using 1% chicken RBCs before use in HAI assays. Serial two-fold dilutions of 25 µL

nanobodies (starting at 300 ng/μL) were prepared in V-bottom microtiter plates. Subsequently, 25 μL of the 4 HA unit virus was added to each well and incubated for 30 min at room temperature (RT). After incubation, 50 μl of 0.5% chicken RBCs was added, and the mixture was further incubated for 30 min at RT. HA inhibition was visually assessed by the formation of well-defined RBC "buttons" or teardrop patterns upon plate tilting. HAI titers were reported as the reciprocal of the highest dilution that completely inhibited hemagglutination.

### Entry, internalization, replication and release assays

For detection of nanobody to block viral entry, H7N9 virus (MOI = 10) was incubated at 37 °C for 1 h with or without nanobody. Then added into A549 cells, incubated at 4 °C for 1 h, thereafter cells were incubated in 37 °C for 1 h before cell collection for RNA extraction.

For detection of nanobody to block viral internalization, H7N9 virus (MOI = 10) was incubated at 37 °C for 1 h with or without nanobody. Then added to A549 cells, incubated at 4 °C for 1 h, then washed with pH=3 PBS three times and incubated 4 h 37 °C. Then cells were collected for RNA extraction.

For detection of nanobody to interfere with viral replication, H7N9 virus (MOI = 10) was incubated at 37 °C for 1 h with or without nanobody. Then added to A549 cells, incubated at 4 °C for 1 h, then washed with pH = 3 PBS three times and incubated 8 h 37 °C. Then cells were collected for RNA extraction.

For testing of nanobody to block viral release, H7N9 virus (MOI = 10) was added to A549 cells at 37 °C for 1 h. Then cells were washed with pH = 3 PBS three times. After 10 h cells were further washed with PBS three times and nanobodies added. After 2 h, cells were collected for RNA extraction.

### RNA isolation and quantitative RT-PCR

Total RNA from cells was extracted with TRIzol following the manufacturer's instructions. For mRNAs, total RNA was transcribed into cDNA using M-MLV Reverse Transcriptase, according to the manufacturer's protocol (Promega, M1701). GAPDH was used as an invariant control for mRNAs. Real-time PCR was carried out using the Light Cycler 480 System (Roche). The RNA level of each gene was shown as the fold of induction ($2^{-\Delta\Delta CT}$) in the graphs. Primer sequences are shown in Table S1.

### Prophylactic and therapeutic nanobody administration in mice

For prophylactic evaluation, groups of 16 mice were intraperitoneally (i.p.) injected with 10 mg/kg purified nanobodies 24 h before being challenged intranasal (i.n.) with 25 μL of virus (H7N9:$10^6$ EID$_{50}$, H3N2:$10^7$ TCID$_{50}$, H1N1: $3 \times 10^3$ TCID$_{50}$) suspended in Hanks' Balanced Salt Solution (HBSS) + 0.1% fetal bovine serum (FBS). Mice were anesthetized with carbon dioxide ice or isoflurane during virus challenge. For therapeutic assessment, mice were infected i.n. with virus (H7N9: $10^6$ EID$_{50}$, H3N2:$10^7$ TCID$_{50}$, H1N1: $3 \times 10^3$ TCID$_{50}$, H7N9-MUT (T166, S167): $10^6$ EID$_{50}$) and treated 2 h later with a single i.p. dose of 25 mg/kg purified nanobodies. Control groups received PBS instead of nanobodies. Mice were monitored daily for clinical signs and body weight loss. On predetermined days post-infection, mice were euthanized, and tissues (nasal turbinate, lung, spleen, kidney, brain, liver) were collected for viral titration and histopathological examination.

### Escape mutation selection

For egg-based selection,100 μL of H7N9 virus ($10^6$ TCID$_{50}$) was incubate at 37 °C for 1 h with 100 μL of nanobodies at varying concentrations (500, 50, 5, 0 μg/mL). The virus-antibody mixture was then injected into 10-day-old SPF (specific-pathogen-free) eggs, and after 40–44 h incubation, the supernatant was collected and tested by hemagglutination assay (HA). The presence of viruses and their escape

capacity was detected by HA and HAI assays. Virus samples positive in both assays were sequenced using next-generation sequencing analysis (NGS) to identify amino acid mutations in the HA segment.

For cell-based selection, MDCK cells were infected with H7N9 virus in the presence of increasing concentration of nanobodies. After 1 h at 37 °C with 5% CO$_2$, the virus containing medium was replaced with virus growth medium (VGM) containing nanobodies, and incubation continued until cytopathic effect (CPE) was observed. Supernatants were collected, and the virus was passaged 10 times, doubling the nanobody concentration after each passage. Viruses from the final passage were sequenced using NGS to identify mutations across all segments.

### Next-generation sequencing (NGS) analysis

Viral RNA was extracted from the egg supernatant and reverse-transcribed into cDNA using the Uni12 primer (5′-AGCRAAAGCAGG-3′). Full-length HA gene amplification was conducted using specific primers (Supplementary Table 1). NGS was performed and analyzed by Tsingke (Tsingke Biotechnology Co., Ltd.).

### Phage display mapping of mAbs

To map the nanobody epitopes, a phage display library was constructed. The full-length H7 protein was divided into 20-mer sequences with 15 amino acid overlaps. The sequences were adjusted to remove unwanted restriction enzyme sites and then synthesized as concatamers each containing ten peptides. The resulting concatamers were then individually cloned into a standard cloning vector. A total of 11 plasmids were produced encoding a total of 109 peptide sequences (ATG: biosynthetics GmbH). The individual plasmids were first transformed into DH5-alpha competent cells by electroporation for maintenance. The 20-mer fragments were then cloned into the pAraPgIII phagemid expression vector. Briefly, the region from each plasmid containing the concatemer sequences was amplified by PCR using Platinum SuperFi II PCR Mastermix (Invitrogen, catalog: 12368010) and universal plasmid-located primers. The amplified DNA was purified (QIAGEN KIT) and the concentration of each amplicon was determined. Amplicons were then pooled such that each fragment was present at the same concentration. The pooled DNA was then digested with PstI and HindIII using FASTDigest enzymes (Thermo Scientific) and dephosphorylated using FASTAP alkaline phosphatase (Thermo Scientific). The digested dephosphorylated fragments were then mixed at a ten-fold molar excess with PstI/HindIII-digested pAraPgIII vector. The mixture was subjected to a ligation reaction with T4 DNA ligase (Thermo Scientific, catalog: EL0014) at room temperature overnight. The ligation reaction was transformed to ElectroMAX DH12S Cells (Invitrogen, catalog: 183812017). Small aliquots of the transformation were spread onto LB agar plates supplemented with ampicillin and incubated overnight at 37 °C to determine the transformation frequency, the remainder was transferred to 25 ml sterile liquid LB broth supplemented with ampicillin and grown overnight at 37 °C with shaking. Aliquots of the resulting culture containing the peptide library cloned into the pAraPgIII vector were used to prepare glycerol stocks that were stored at −80 °C. Further aliquots were used for plasmid purification.

Cloning efficiency was determined by restriction enzyme analysis of purified plasmid. Plasmid preparations were digested with KpnI/NotI. Empty vector digest with both enzymes whereas plasmids carrying insert digests with KpnI only. The library was found to contain less than 5% empty plasmid. NGS Illumina sequencing was used to demonstrate that all the peptides were present in the library.

To express the library on the surface of M13 phages the M13KO7 helper phage (supplied by Thermo Fisher) was used. Two aliquots of 5 mL of terrific broth (GIFCO, 009600) supplemented with 100 μg/ml ampicillin (without glucose) were inoculated with 50 μl of an overnight starter culture, followed by the addition of M13K07 helper phage at a

multiplicity of infection of aproximately100:1. After 2.5 h, 50 µg/ml kanamycin was added to one of the cultures (as a non-induced control) and kanamycin and 0.2% arabinose was added to the second culture (to express the peptides on the phage surface). The culture was incubated at 37 °C with shaking (180 rpm) overnight. Next days, cells were removed from the supernatant by centrifugation followed by filtration through a 0.2 µM filter. The number of phagemid carrying particles in the resulting supernatant were determined by infecting TOP10 F' *E. coli* cells with serially diluted phage library suspensions and incubated for 90 min at 37 °C, 180 rpm. A total of 100 µl of each dilution was plated on Lubia broth agar plates supplemented with 100 µg/ml ampicillin and incubated overnight at 37 °C. The number of colonies were counted the next day.

A panning experiment was performed to check which epitopes the nanobodies bind to. Ninety-six-well NUNC plates (Thermo Fisher, 10394751) were coated with 25 ug of each nanobody, at 125 µl/well in triplicate, in 0.1 M bicarbonate buffer. Plates were incubated overnight at 4 °C. The next day the coating solution was removed. The wells were washed three times with 0.1 M bicarbonate buffer and blocked for 1 h at 4 °C with 300 ul/well of filter sterilized 0.1 M bicarbonate buffer containing 0.5% BSA. After blocking, 100 µl of phage suspension containing $3.9 \times 10^9$ phagemid-carrying particles/ml was added to each well and incubated for 1 h at room temperature, with gentle rocking. Wells were then washed 8 times with 50 mM Tris-HCL pH 7.5, 150 mM NaCl and 0.1% Tween-20 to remove unbound phages. Bound phages were then eluted by adding 100 µl/well of 0.2 M Glycine HCl pH2.2 with 0.1% BSA and incubating for 12 min at room temperature. Fifteen µl of 1 M Tris buffer pH 9.1 was added to neutralize the solution. The eluted phage suspension was then stored at 4 °C.

To determine the epitopes bound by the nanobodies, eluted phages were used to infect TOP10 F' *E. coli* as described above. Following infection, the cells were spread onto LB agar plates supplemented with ampicillin and incubated overnight at 37 °C. Plasmids were isolated from individual colonies and sent for sequencing (Eurofins Scientific) using an araC universal primer. Results from the sequencing were analyzed using SnapGene and mapped on a 3D structure using Open-Source PyMOL version 2.5.0.

## Viral growth kinetics

Triplicate wells of confluent MDCK cells were infected with WT and MUT virus at a MOI = 0.001 and incubated with BSA-MEM containing 1 µg/ml TPCK-treated trypsin at 37 °C. Supernatants were harvested at 6-, 12-, 24-, 36 h post-infection and titrated by $TCID_{50}$ in quadruplicate on MDCK cells in a 96-well plates. After 48 h, presence of virus in the wells was determined by hemagglutination using 1% chicken Red Blood Cells. $TCID_{50}$ of the samples was calculated using the Reed and Muench method.

## Rescue of H7N1 virus

The H7N1-WT and H7N1-MUT viruses were rescued under BSL-2 conditions, as viruses containing HA of Influenza A/Environment/Suzhou/SZ19/2014 and the backbone segments of A/Puerto Rico/8/1934/H1N1 (PR8 strain). Influenza A reverse genetics was performed using a previously established protocol[80] to rescue the H7N1-WT and H7N1-MUT viruses. Briefly, plasmid cocktails were prepared to contain pDZ plasmids of the seven PR8 backbone segments with either the pDZ H7 SZ19-WT or the pDZ H7 SZ19-MUT plasmid respectively. Co-cultures of HEK293T and MDCK-SIAT1 cells were reverse transfected with the respective plasmid cocktails using TransitLT1 reagent (Mirus Bio, lyec-1) in a 1:4 ratio. Monolayers of MDCK-SIAT1 cells were then infected with the co-culture supernatant and observed for cytopathic effect after 72 h. The virus rescues were confirmed using hemagglutination assay and their H7 ORF sequences were confirmed using Sanger sequencing.

## HA sequence analysis and alignment

To determine the distributions of amino acids at position 166,167 of HA in influenza A virus derived from different species, a total of HA amino acid sequences was downloaded from the Global Initiative on Sharing All Influenza Data (GISAID) (https://www.gisaid.org/) database and the Influenza Virus Database of GenBank (https://www.ncbi.nlm.nih.gov/genomes/FLU/Database/nph-select.cgi?go_database). The amino acid sequences of each HA were aligned using MAFFT[81]. Base compositional data of the amino acid at position 166,167 were then graphically plotted using the Python language.

## Virus infection of mice and flow cytometry analysis

The rescued virus H7N1 (WT) and H7N1-T166, L167 (MUT) were generated following previous protocols. C57BL/6 mice, maintained in our laboratory in Gothenburg, were infected intranasally with either WT virus (0.1 $TCID_{50}$) or MUT virus (50 $TCID_{50}$) per mouse in Hanks' Balanced Salt Solution (HBSS) + 0.1% fetal calf serum (BSA) intranasally in a volume of 25 µL/mouse. Fourteen days post-infection, the mice were sacrificed and mediastinal lymph nodes (medLNs), spleens and lungs harvested for further analysis. The harvested organs were processed to obtain a single-cell suspension, and red blood cells were lysed using ACK lysing buffer. For B cell characterization, 25 µL of premixed extracellular antibody mixture was added to each sample and incubated for 30 min at 4 °C. The following antibodies were used for extracellular B cell staining: 1:700 anti-mouse CD3 BV510 (BD, catalog: 563024), 1:200 anti-mouse IgM PE/Dazzle 594 (BioLegend, catalog: 314529), 1:200 anti-mouse IgD BV785 (BD, catalog: 563618), 1:200 anti-mouse CD38 FITC (BioLegend, catalog: 102705), 1:200 anti-mouse GL7 PE (BioLegend, catalog: 144607), and 1:200 anti-mouse B220 Pe-Cy7 (BioLegend, catalog: 103221). Following the antibody incubation, the samples were washed with 1 mL of FACS-buffer (DPBS + 2% FBS + 0.5 mM EDTA), and the supernatant was discarded. To assess HA-specific B cells, 100 µL of premixed WT/MUT HA proteins were added to each tube and incubated for 1 h at 4 °C. WT-HA protein was conjugated with streptavidin-APC (Bioligand, catalog: 405243), Brilliant Violet 421 Streptavidin (BioLegend, catalog: 405225), MUT-HA protein was conjugated with Brilliant Violet 650 Streptavidin (BD, catalog: 563855), Percp5.5 Streptavidin (BioLegend, catalog: 405214) before the experiment: proteins were diluted at 0.02 mg/ml and labeled by stepwise addition (at 10 min intervals) of molar excess of fluorescent streptavidin. Labeled proteins were stored at 4 °C overnight and used within 2 weeks of labeling[18]. The samples were then washed. To exclude dead cells, Live/Dead Aqua (Invitrogen, catalog: L34966) staining was performed for 30 min at 4 °C, followed by fixation of the cells with 1.5% paraformaldehyde (PFA). After fixation, the samples were washed, resuspended in 200 µl of FACS buffer, and stored at 4 °C until analysis.

## B cell ELISpot assay

Ninety-six-well ELISpot plates (Millipore Sigma) were coated with 100 µL per well of recombinant WT-HA protein or MUT-HA at a concentration of 1 µg/mL, and incubated overnight at 4 °C. The next day, the plates were blocked with 100 µL of PBS containing 2% BSA and incubated for at least 1 h at RT. Mice were sacrificed 14 days after being infected with WT (0.1 $TCID_{50}$) /MUT (50 $TCID_{50}$) virus 14 days, and their medLNs were harvested and processed for analysis. Cells isolated from medLNs were counted using a Muse cell analyzer (Merck Millipore) in 0.1% BSA in PBS. Each well was seeded with $1.5 \times 105$ cell in 150 µL of DMEM supplemented with 10% FBS and 10 µg/mL gentamicin, then serially diluted 3-fold and incubated overnight at 37 °C. Plates were washed three times with PBS containing 0.05% Tween, followed by a 2 h incubation with 50 µl anti-mouse IgG H + L HRP (Aviva Systems Biology, catalog: ORA 04973) at RT. After three additional washes, the plates were developed using BD ELISpot AEC Substrate solution and incubated in a humid chamber for 10 min at RT before stopping the

reaction. A CTL ImmunoSpot plate reader was used for imaging, and spots were manually counted. The number of spot-forming cells was normalized to $10^6$ cells.

## Atomic model building and refinement

For structure determination, a model of nanobodies was generated using ImmuneBuilder[54]. The model of nanobodies linked with Fc tag was done using the ColabFold version of AlphaFold v2.3.0[82,83], SZ19-HA and SZ19-HA-$_{T166,L167}$ were generated by Swiss-model (https://swissmodel.expasy.org/). Structures were analyzed and figures were generated using Open-Source PyMOL version 2.5.0. (http://www.pymol.org). Final model statistics were summarized in Supplementary Fig. 1B. The ClusPro 2.0[84,85] server was used in "antibody mode" to produce docking models between nanobody E10-Fc and SZ19 H7-HA. The top 30 models returned by the server were ranked depending on energy and cluster size. The 10 best ClusPro docking models were further analyzed based on interactions and interface properties calculated according to PDBePISA (Proteins, Interfaces, Structures and Assemblies) (European Bioinformatics Institute (https://www.ebi.ac.uk/pdbe/prot_int/pistart.html))[86] and combined in our primarily results in Fig. 4a. The final model subsequently underwent manual analysis and image generation in Open-Source PyMOL version 2.5.0.

## Peptide preparation and immunization

H7-HA$_{166-186}$ peptide (length: 21 amino acids, sequence KSYKNTRK-SPAIIVWGIHHSV) was linked with OVA protein at the C-Terminal. Mice were immunized with H7-HA$_{166-186}$/OVA peptide or just PBS three times with i.p. The first two times each mouse immunized intraperitoneally with 100 μg H7-HA$_{166-186}$/OVA peptides in 100 μL PBS mixed with 100 μL of AddaSO3™ Adjuvant (InvivoGen, vac-as03-10). For the third immunization the same amount of peptide was used, but without adjuvant. The H7-HA$_{166-186}$ peptide was synthesis by and purchased from GenScript and the purity is ≥95%.

## Statistical analysis

GraphPad Prism software was used for plotting and statistical analysis. To compare the counts of cells between the groups in the ELISpot screening analysis, a one-way ANOVA was employed, followed by Dunnett's multiple comparisons test to adjust for comparisons between individual pools and the control peptide. For nanobody detection, a two-sided unpaired Student's $t$ test was performed. Statistical significance was indicated as follows: **** for $p$ value ≤ 0.0001, *** for $p$ value ≤ 0.001, ** for $p$ value ≤ 0.01, and * for $p$ value ≤ 0.05. Additionally, a two-way ANOVA with post hoc Tukey's HSD test was performed, using an alpha level of 0.05 to identify significant differences across multiple factors. Statistical significance for these analyses was similarly indicated with **** for $p$ value of ≤0.0001 and * denotes a $p$ value of ≤0.05.

## Reporting summary

Further information on research design is available in the Nature Portfolio Reporting Summary linked to this article.

## Data availability

All data are available in the main text or the Supplementary Materials. Source data are provided with this paper.

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

## Acknowledgements

We would like to thank all members of the Angeletti lab for helpful discussions regarding the project. We thank Zhengxiang Wang for the help on the quantify of virus EID$_{50}$. We would like to acknowledge Protein Production Sweden for provisioning of facilities and experimental support, and we would like to thank M. Andersson and M. Bäckström for assistance. Protein Production Sweden is funded by the Swedish Research Council as a national research infrastructure. We thank the laboratory for Experimental Biomedicine (EBM) at the University of Gothenburg for assistance with mouse breeding and husbandry. The study was supported by grants from the National Key R&D program (grant no. 2021YFD1800204 to Q.Z.), the National Natural Science Foundation of China (grant nos. U23A20243 and 32272972 to Q.Z.), the Science and Technology Major Project of Gansu Province (grant nos. 23JRRA1513 and 24JRRA806 to Q.Z.), the Swedish Society for Medical Research (grant no. S19-0019 to A.A.S.), the European Research Council (ERC-StG, B-DOMINANCE, grant no. 850638 to D.A.); the Swedish Research Council (grant no. 2019-01855 to A.A.S. and grant nos. 2021-01164 and 2021-01165 to D.A.); the Knut and Alice Wallenberg Foundation (grant no. PAR 2020/228 to A.A.S. and 2021.0033 to D.A.).

## Author contributions

Conceptualization/Project administration: Q.Z. and D.A. Methodology: Z.-S.C., H.-C.H., M.L., Y.Ji, Q.Z., and D.A. Investigation: Z.-S.C., X.W., K.S., Y.Jia, J.R.M., and D.F.B. Formal analysis: Z.-S.C., H.-C.H., X.W., Q.Z., and D.A. Writing—original draft: Z.-S.C., H.-C.H., A.A.S., and D.A. Writing—review and editing: all authors. Technical support: A.A.S. and G.D. Supervision: Q.Z. and D.A. Funding acquisition: Q.Z. and D.A.

## Funding

## Competing interests

The authors declare no competing interests.
