## [Transparent Peer Review file · Nature Communications]

Influenza A Virus H7 nanobody recognizes a conserved immunodominant epitope on hemagglutinin head and confers heterosubtypic protection

Corresponding Author: Dr Davide Angeletti

Version 0:

Reviewer comments:

Reviewer #1

(Remarks to the Author)

The authors immunized a male 2.5-year-old Alpaca with inactivated H7N9 SZ19/2014 five times and isolated six H7-reactive nanobodies (VHH) from PBMCs; one of them – namely, E10, was neutralizing, protective, and cross-reactive against H7N9, H3N2, and H1N1, with epitope mapped to a linear peptide in the lateral patch of HA head involving the key residues K166 and S167 and nearby two beta-strands. The major findings are: 1) the nanobody E10 and its epitope mapped through viral escape mutations and phage display of overlapping peptides; 2) the E10 escape mutant (K166T and S167L) was less fit; 3) the E10 epitope was immunodominant in mice infected with H7N9 SZ19/2014; 4) H7HA166-186 peptide immunization in mice conferred partial protection. The authors conclude that E10 is a promising therapeutic for H7N9 and the E10-epitope is a potential target for cross-subtype or cross-group influenza vaccines. Overall, the data supported the major findings and conclusions. However, the authors should comment on the E10 concentration used for ELISA and neutralization, i.e., 100 ng/μL or 100 μg/mL, and 10 mg/kg for mice prophylactic, which are 10 times higher than a typical neutralizing IgG1 mAb. Also, E10 is clearly more reactive to H7 than H3 or H1 – this critical difference should be stated throughout the manuscript.

1. Lines 65-66, “Among those” was an incomplete sentence.
2. The Alpaca immunization route was missing.
3. Line 98, immunized with “inactivated” H7N9 virus
4. Line 109, the longest CDR3 “of 17 amino acids”
5. Lines 111-112, at different points after “SZ19” infection “of A549 cells”
6. Lines 113-114, SZ19 virus-infected “A549” cells
7. Fig 1E, the 50% neutralization endpoint should be nanobody concentration, not sample dilution – or indicate the starting concentration of 100 ng/μL for sample dilutions shown.
8. Fig 1G, in the table, what’s the unit for nanobody concentration? In the plot, HAI titer should be nanobody concentration – or indicate the starting concentration of 100 ng/μL for sample dilutions shown.
9. Fig 2G, the 50% neutralization endpoint should be E10 concentration, not sample dilution – or indicate the starting concentration of 100 ng/μL for sample dilutions shown.
10. Fig 2H, signals for E10 seemed to be in opposite directions.
11. Line 177, E10 administration route was missing.
12. Line 197, although H3N2 and H1N1 are IAV subtypes “currently circulating in humans”, the strains tested – X31 for H3N2 and PR8 for H1N1 – are not contemporary circulating strains. This should be clearly stated.
13. Lines 235-237, Fig 4F, the description and presentation of the structural modeling were poor.
14. Line 261, cite Fig S4E, which showed prophylactic data.
15. Fig 6E, Fig S5D-E, what was “% of parent”?
16. Fig S6, serum dilution seemed to be in reverse direction.
17. Line 367, the H7 lateral patch human mAbs could not bind H10 and H12.
18. Line 377, “KPF1” came from nowhere.

Reviewer #2

(Remarks to the Author)

Chen et al report on a H7 nanobody specific for a conserved epitope on hemagglutinin head and its protective potential. The study is a useful addition to the field both in terms of developing countermeasures against influenza viruses as well as from an immunological perspective.

The following points should be addressed prior to publication:

1. I have concerns about the validity of the HAI data. Line 132 "Surprisingly, even the non-neutralizing nanobody H12 demonstrated HI activity" – please explain how this works? Along similar lines, nanobody F3 was mapped to the HA stem by phage display (Fig 4C and S3) but shows HAI activity (comparable to the head-specific A11 in Fig 1).
2. Line 152 "and it demonstrated effective neutralization of H1N1 and H3N2 (Fig. 2G)" – the potency against H1 and H3 seems to be relatively low compared to that against H7 (Fig 1E). It would be worth plotting the three viruses on the same figure for comparison. This is pertinent as it links to the lower protective efficacy against H1 and H3 seen in mice and the lower binding seen in Fig 2.
3. The reporting of neutralisation and HI assays is somewhat unclear. Please clarify the points below.
 - a. Figure 1E legend reads "Graph showing 50% neutralization endpoint (NC50) of different nanobodies against the SZ19 as measured by their ability to inhibit hemagglutination of red blood cells (RBCs)." This is unclear. If this is the neutralisation assay with the RBC readout, then the assay is not testing if the nanobodies inhibit hemagglutination of RBCs.
 - b. Line 553 – "Hemagglutination assay" in methods, the first section describes a neutralisation assay with HA readout, not a Hemagglutination assay itself. This is partly described, and should be under the 'Neutralisation assay' section.
 - c. line 567 "For nanobody attachment detection, H7N9 virus (MOI=10) was incubated at 37 °C for 1h. Then added into A549 cells and kept in 4 °C for 1h, and put in 37 °C 1h, then collected cells for RNA extraction." Why is the virus incubated at 37 for 1hr? Is that in the presence of the nanobody? How does detection of RNA in the cells demonstrate inhibition of attachment?
4. The authors may want to tone down the 'remarkable' features of E10. While this is indeed a broadly binding nanobody, its cross-reactivity potency is not that 'remarkable'. As mentioned above, it is only weakly neutralising against H1 and H3, and it reduces mortality in mice, there is still considerable weight loss observed. Similarly, the comparisons with the previously published H7 lateral patch mAbs is not appropriate as the Jia et al 2024 study only tested their mAbs up to 10ug/ml while the current study tests much higher concentrations of their nanobodies and only sees H1 or H3 cross-binding at concentrations higher than 10ug/ml.

Version 1:

Reviewer comments:

Reviewer #1

(Remarks to the Author)

The authors addressed the previous critiques well.

A few more clarifications and corrections:

Line 374, Fab6649 is the same mAb as CL6649.

Lines 484-486, the amounts of Freund's Complete and Incomplete Adjuvants were missing.

Line 591, nanobody "to block viral" entry

Line 594, nanobody "to block viral" internalization

Line 598, nanobody "to interfere with viral" replication

Line 602, For testing of nanobody "to block viral" release

Reviewer #2

(Remarks to the Author)

The authors have addressed my comments

We would like to thank both reviewers for the positive assessment of our work. We have carefully addressed all of their comments and made the appropriate modifications within the manuscript. In addition, we have gone through the manuscript and improved the method section.

Please find below a point-by-point response.

Reviewer #1 (Remarks to the Author):

The authors immunized a male 2.5-year-old Alpaca with inactivated H7N9 SZ19/2014 five times and isolated six H7-reactive nanobodies (VHH) from PBMCs; one of them – namely, E10, was neutralizing, protective, and cross-reactive against H7N9, H3N2, and H1N1, with epitope mapped to a linear peptide in the lateral patch of HA head involving the key residues K166 and S167 and nearby two beta-strands. The major findings are: 1) the nanobody E10 and its epitope mapped through viral escape mutations and phage display of overlapping peptides; 2) the E10 escape mutant (K166T and S167L) was less fit; 3) the E10 epitope was immunodominant in mice infected with H7N9 SZ19/2014; 4) H7HA166-186 peptide immunization in mice conferred partial protection. The authors conclude that E10 is a promising therapeutic for H7N9 and the E10-epitope is a potential target for cross-subtype or cross-group influenza vaccines. Overall, the data supported the major findings and conclusions. However, the authors should comment on the E10 concentration used for ELISA and neutralization, i.e., 100 ng/μL or 100 μg/mL, and 10 mg/kg for mice prophylactic, which are 10 times higher than a typical neutralizing IgG1 mAb. Also, E10 is clearly more reactive to H7 than H3 or H1 – this critical difference should be stated throughout the manuscript.

We thank the reviewer for their positive comments regarding our work. We agree with their assessment that the ELISA and neutralization concentrations, that bind to and have an effect on H1 and H3, are higher than what typically is seen for IgG1 mAbs. On the other hand, 10mg/kg in mice is on the higher side, but still well within the range of what is used in other studies (see for example Yu et al., 2017, Cell Host & Microbe; Dong et al., 2020, JCI; Gilchuk et al., 2019, Cell Host & Microbe; etc). Nevertheless, the binding is specific, as no other nanobody demonstrates cross-reactivity. We have now clearly stated that the dose is higher than mAbs and that binding and protection remains better against H7, as compared to H1 and H3 (Discussion, line 378 and following). We also discuss that the nanobody may be further improved to increase its affinity for H1 and H3.

1. Lines 65-66, “Among those” was an incomplete sentence.

We rewrote the sentence

2. The Alpaca immunization route was missing.

The alpaca was immunized intramuscularly. We now specify the immunization route

3. Line 98, immunized with “inactivated” H7N9 virus

We corrected the sentence

4. Line 109, the longest CDR3 “of 17 amino acids”

We corrected the sentence

5. Lines 111-112, at different points after “SZ19” infection “of A549 cells”

We corrected the sentence

6. Lines 113-114, SZ19 virus-infected “A549” cells

We corrected the sentence

7. Fig 1E, the 50% neutralization endpoint should be nanobody concentration, not sample dilution – or indicate the starting concentration of 100 ng/μL for sample dilutions shown.

The reviewer is correct. We have now corrected the figure and indicate the concentration of nanobody showing 50% neutralization titer.

8. Fig 1G, in the table, what’s the unit for nanobody concentration? In the plot, HAI titer should be nanobody concentration – or indicate the starting concentration of 100 ng/μL for sample dilutions shown.

The reviewer is correct. In the table the concentration was already ug/ml. We have now clearly indicated this. In addition we have changed the graph to represent the concentration of nanobody.

9. Fig 2G, the 50% neutralization endpoint should be E10 concentration, not sample dilution – or indicate the starting concentration of 100 ng/μL for sample dilutions shown.

Figure 2G has been revised to reflect the concentration. Furthermore, as suggested by reviewer #2, we have added E10 neutralization of H7N9

10. Fig 2H, signals for E10 seemed to be in opposite directions.

We thank the reviewer for noticing this. We have now fixed the figure

11. Line 177, E10 administration route was missing.

We have now specified the administration route

12. Line 197, although H3N2 and H1N1 are IAV subtypes “currently circulating in humans”, the strains tested – X31 for H3N2 and PR8 for H1N1 – are not contemporary circulating strains. This should be clearly stated.

The reviewer is correct. We have amended the sentence removing the “currently circulating in humans” part and clearly wrote in the discussion that these strains are not contemporary (line 431 and following).

13. Lines 235-237, Fig 4F, the description and presentation of the structural modeling were poor.

We have redone the figure to improve the resolution. Furthermore, we have amended the text (method and figure legend) to improve clarity

14. Line 261, cite Fig S4E, which showed prophylactic data.

We added the figure citation

15. Fig 6E, Fig S5D-E, what was “% of parent”?

% of parent, indicated that the data shown was the frequency of the parent gate. For clarity we have changed to “% of GC/MBC”

16. Fig S6, serum dilution seemed to be in reverse direction.

We thank the reviewer for spotting this, we have reversed the dilution in the correct orientation for Fig S5G and S6

17. Line 367, the H7 lateral patch human mAbs could not bind H10 and H12.

We apologize for the oversight, we have amended the sentence and removed the reference to H10 and H12 binding

18. Line 377, “KPF1” came from nowhere.

KPF1 was the mAb identified in the cited study. But we agree with the reviewer that it was confusing, we have therefore removed the mAb name

Reviewer #2 (Remarks to the Author):

Chen et al report on a H7 nanobody specific for a conserved epitope on hemagglutinin head and its protective potential. The study is a useful addition to the field both in terms of developing countermeasures against influenza viruses as well as from an immunological perspective.

The following points should be addressed prior to publication:

We thank the reviewer for the positive evaluation of our work

1. I have concerns about the validity of the HAI data. Line 132 “Surprisingly, even the non-neutralizing nanobody H12 demonstrated HI activity” – please explain how this works? Along similar lines, nanobody F3 was mapped to the HA stem by phage display (Fig 4C and S3) but shows HAI activity (comparable to the head-specific A11 in Fig 1).

We have repeated the HAI assay and obtained the same results. Please note that the experiment has been performed both in the laboratory in Sweden as well as in the laboratory in China by two independent people (See below **Fig 1 for reviewers**). However, the reviewer is correct, that the H12 result is extremely puzzling. Indeed, H12 also selected for escape mutants (fig S3A) so we believe the neutralization assay we used in Fig 1F is maybe not sensitive enough. Some weak neutralization can be detected from the assay performed in Fig 1E. Nevertheless, we looked into the literature and found some reports (of polyclonal sera) where sera was high in HAI but negative in MN (Verschoor et al., Plos One 2015 – see Fig 1, pasted below as **Fig 2 for reviewers**; Truelove et al., influenza other respir viruses 2016 – see fig 2, pasted below as **Fig 2 for reviewers**).

While we do not have a conclusive explanation, the two assays are different and use cells from different species (Avian vs human), which have different kind of sialic acid and also have different receptor density on their surface. How many HA-sialic acid interactions need to be disrupted in order to inhibit hemagglutination? And what is the relative avidity of H7-

HA for sialic acid on human cells vs avidity for the nanobody? These are all factors that could play a role in the discrepancy observed.

Overall, we can remove the HAI figure, if the reviewer and the editor wish. Since the analyses of nanobodies (other than E10) is not central to the study we think it will be fine to leave it and we reserve to further delve into their mechanisms of action in future studies. For now we have added further discussions in line 136 and 232

Fig 1 for reviewers. Old Fig 1G (with data converted to ug/ml, as requested by reviewer #1) compared with new experiment performed in response to this comment. The trend repeats yet once again showing robustness of the assay. Please note that HAI assay is non quantitative so one cannot compare the amount obtained in the first repeat vs second repeat but one can compare between samples.

Fig 2 for reviewers. On the left is Fig 1 from Verschoor et al., Plos One 2015; on the right is Fig 2 from Truelove et al., influenza other respir viruses 2016. Both studies find polyclonal Abs that block HAI but do not neutralize.

2. Line 152 “and it demonstrated effective neutralization of H1N1 and H3N2 (Fig. 2G)” – the potency against H1 and H3 seems to be relatively low compared to that against H7 (Fig 1E). It would be worth plotting the three viruses on the same figure for comparison. This is pertinent as it links to the lower protective efficacy against H1 and H3 seen in mice and the lower binding seen in Fig 2.

We agree that the relative potency to H3 and H1 is lower as compared to H7. This is to be expected as H7 is the original virus against which the nanobody was raised. We have added H7 neutralization to Fig 2G and clearly stated this observation in the text

3. The reporting of neutralisation and HI assays is somewhat unclear. Please clarify the points below.

We have now addressed the points below and further clarified how we performed the assays within the text and methods

a. Figure 1E legend reads “Graph showing 50% neutralization endpoint (NC50) of different nanobodies against the SZ19 as measured by their ability to inhibit hemagglutination of red blood cells (RBCs).” This is unclear. If this is the neutralisation assay with the RBC readout, then the assay is not testing if the nanobodies inhibit hemagglutination of RBCs.

The reviewer is correct, indeed there was an error in the legend. The legend has been amended and read as “Graph showing 50% neutralization endpoint (NC50) of different nanobodies against the SZ19 H7N9 virus. Virus was incubated with serially diluted nanobodies before cell infection. After 72h, viral replication was measured by the ability of supernatant to hemagglutinate red blood cells (RBCs).”

b. Line 553 – “Hemagglutination assay” in methods, the first section describes a neutralisation assay with HA readout, not a Hemagglutination assay itself. This is partly described, and should be under the ‘Neutralisation assay’ section.

Again, the reviewer is correct, we have carefully revised the whole method section and better described the different methods used

c. line 567 “For nanobody attachment detection, H7N9 virus (MOI=10) was incubated at 37 °C for 1h. Then added into A549 cells and kept in 4 °C for 1h, and put in 37 °C 1h, then collected cells for RNA extraction.” Why is the virus incubated at 37 for 1hr? is that in the presence of the nanobody? How does detection of RNA in the cells demonstrate inhibition of attachment?

We have amended the method to better explain the experimental setup. Here virus was incubated at 37C either with or without nanobody. Thereafter, the cells were placed at 4C for 1h, followed by 1h at 37C. This assay was aimed at detecting inhibition of both attachment and entry: if RNA was present within cells then the virus must have attached to receptor and entered the cells. As we are not able to fully distinguish the two (attachment and entry) and as attachment is also measured by HAI assay, we have changed this to “Viral Entry”

4. The authors may want to tone down the ‘remarkable’ features of E10. While this is indeed a broadly binding nanobody, its cross-reactivity potency is not that ‘remarkable’. As mentioned above, it is only weakly neutralising against H1 and H3, and it reduces mortality in mice, there is still considerable weight loss observed. Similarly, the comparisons with the previously published H7 lateral patch mAbs is not appropriate as the Jia et al 2024 study only tested their mAbs up to 10ug/ml while the current study tests much higher concentrations of their nanobodies and only sees H1 or H3 cross-binding at concentrations higher than 10ug/ml.

We have changed the tone of the manuscript throughout. As discussed in our response to reviewer #1, we now clearly acknowledge the limitations of our nanobody and the fact that a higher dose is needed for cross-binding and neutralization.

We thank again the reviewers for taking the time to go through our study.

Please find below the response to the outstanding comments

Reviewer #1 (Remarks to the Author):

The authors addressed the previous critiques well.

A few more clarifications and corrections:

Line 374, Fab6649 is the same mAb as CL6649.

We have removed the sentence about Fab6649

Lines 484-486, the amounts of Freund's Complete and Incomplete Adjuvants were missing.

Clarified the method regarding alpaca immunization. Now it reads as “The first immunization used 300µg of antigen in 300µl PBS + 700µl Freund's Complete Adjuvant (FCA, Thermo, 77140) followed by four subsequent immunizations with 200µg of antigen in 200µl PBS + 800µl Freund's Incomplete Adjuvant (Thermo, 77145).

Line 591, nanobody "to block viral" entry

Line 594, nanobody "to block viral" internalization

Line 598, nanobody "to interfere with viral" replication

Line 602, For testing of nanobody "to block viral" release

Thank you. We have corrected the sentences as suggested above

Reviewer #2 (Remarks to the Author):

The authors have addressed my comments